# Context-Aware Estimation of Attribution Robustness In Text

## Abstract

Explanations are crucial parts of deep neural network (DNN) classifiers. In high stakes applications, faithful and robust explanations are important to understand DNN classifiers and gain trust. However, recent work has shown that state-of-the-art attribution methods in text classifiers are susceptible to imperceptible adversarial perturbations that alter explanations significantly while maintaining the correct prediction outcome. If undetected, this can critically mislead the users of DNNs. Thus, it is crucial to understand the influence of such adversarial perturbations on the networks' explanations. In this work, we establish a novel definition of attribution robustness (AR) in text classification. Crucially, it reflects both attribution change induced by adversarial input alterations and perceptibility of such alterations. Moreover, we introduce a set of measures to effectively capture several aspects of perceptibility of perturbations in text, such as semantic distance to the original text, smoothness and grammaticality of the adversarial samples. We then propose our novel CONTEXT-AWAREEXPLANATIONATTACK (CEA), a strong adversary that provides a tight estimation for attribution robustness in text classification. CEA uses context-aware masked language models to extract word substitutions that result in fluent adversarial samples. Finally, with experiments on several classification architectures, we show that CEA consistently outperforms current state-of-the-art AR estimators, yielding perturbations that alter explanations to a greater extent while being less perceptible.

## 1 Introduction

Attribution methods aim to give insights into causal relationships between deep neural networks' (DNNs) inputs and their outcome prediction. They are fundamental to unravel the black-box nature of DNNs and are widely used both in the image and natural language domain. Commonly used attributions like Saliency Maps (Simonyan et al., 2013), Integrated Gradients (Sundararajan et al., 2017), DeepLIFT (Shrikumar et al., 2017) and Self-Attention (Bahdanau et al., 2015) highlight input features that are deemed important for the DNNs in the inference process. These methods are especially attractive and useful, as they provide on-the-fly explanations without requiring any domain-specific knowledge from users or extensive computation resources.

However, it has been shown recently that many of these attributions lack robustness towards adversarial perturbations (Ghorbani et al., 2019). Carefully crafted, *imperceptible* input alterations change the explanations significantly without modifying the output prediction of the DNNs. This violates the *prediction assumption* of faithful explanations (Jacovi & Goldberg, 2020), which states that similar inputs should have similar explanations for identical outputs. Figure 1 exemplifies this fragility of attributions. In many safety-critical natural language processing problems, such as EHR classification (Girardi et al., 2018), robustness is a key factor for DNNs to be deployed in real life. For instance, a medical professional assessing EHRs would neither understand nor trust a model that yields two significantly different explanations for seemingly identical input texts and predictions. Hence, it is fundamental to understand how the networks and attributions behave in the presence of input perturbations and how perceptible those alterations are to the user.

In this work, we focus on understanding the adversarial robustness of attribution maps (AR) in *text classification* problems. Specifically, we are interested in investigating and quantifying the extent to which small input perturbations can alter explanations in DNNs and how perceptible such alterations are. We do so by

| Original sample | CEA perturbed sample (ours) | TEF perturbed sample (Ivankay et al., 2022) |
|---|---|---|
| **press** the **delete** **key** .

F($s$, "Negative") = 0.99 | hit the delete **key** .
F($s$, "Negative") = 0.95
**r**: 30
*SemS*: 0.98
*PCC*: -0.05 | newspaper **the delete key .**
F($s$, "Negative") = 0.95
**r**: 1.1
*SemS*: 0.8
*PCC*: 0.6 |
| **peek** at the week : **ben** vs. the **streak** **\|** **yet** another risky **game** for that patriots winning streak , now at 21 . pittsburgh hasn # 39;t lost at home , and **rookie** **quarterback** ben roethlisberger hasn # 39;t lost , period .

F($s$, "Sports") = 0.99 | **peek** at the playoffs : ben vs. the **steelers** **\|** **yet** another risky **game** for that patriots winning **streak** , now at 21 **.** **pittsburgh** hasn # **34** lost at home , and rookie **quarterback** **ben** roethlisberger hasn # 39;t lost , period $\geq$

F($s$, "Sports") = 0.95
**r**: 14.9
*SemS*: 0.97
*PCC*: 0.02 | hoodwink at **the** **zou** : **suis** **vs.** the **wave** **\|** **yet** another **risky** game **for** that **patriots** winning **streak** , now **at** 21 **.** **pittsburgh** **hasn** # 39;t lost at **home** , **and** **rookie quarterback** **ben** roethlisberger hasn # 39;t lost , period .
F($s$, "Sports") = 1.0
**r**: 3.4
*SemS*: 0.9
*PCC*: 0.22 |
| intel seen **readying** new **wi** - **fi** chips \| intel corp . this week isexpected to introduce a chip that adds support for a relativelyobscure version of wi - **fi** , analysts said on monday , in a movethat could help ease congestion on **wireless** networks .

F($s$, "Sci/Tech") = 0.78 | intel seen **readying** wireless wi - fi chips \| intel corp . this week isexpected to launch a specification that added support for a relativelyobscure version of wi - fi , analysts said on monday , in a movethat could help ease congestion on wireless networks .

F($s$, "Sci/Tech") = 0.95
**r**: 20
*SemS*: 0.98
*PCC*: 0.27 | intel seen readying nouveau **wi** - **fi** chips \| intel corp . this week isexpected to **insert** a dies that summing support for a relativelyobscure version of wi - **fi** , **analysts** said on monday , in a movethat could help ease congestion on **wireless** networks .

F($s$, "Sci/Tech") = 0.95
**r**: 4
*SemS*: 0.91
*PCC*: 0.28 |

Figure 1: Three examples of fragile attribution maps in text sequence classifiers. In each row, careful alteration of the original sample results in significantly different attribution maps while maintaining the prediction confidence F in the correctly predicted class. Red words have positive attribution values, i.e. contribute *towards* the true class, while blue words with negative attributions *against* it. Our novel CEA attack yields perturbed samples that have lower *Pearson Correlation Coefficient* (PCC) values between the words highlighted by the attribution method in the original and perturbed inputs, as well as higher semantic similarity values (SemS) of the original and adversarial sentences, compared to the baseline TEF attack. This results in higher estimated robustness constants $r$ (see Section 4), thus lower robustness of the classifiers against attacks.

focusing on methods to find perturbations that maximize the change in attribution while being as imperceptible as possible. Characterizing and quantifying the robustness of attribution methods is an important step towards training robust classifiers and attribution methods that can be deployed in a wide variety of critical real-life use cases. We summarize our contributions as follows:

- We are the first to introduce a definition of attribution robustness (AR) in text classification that takes both the attribution distance and perceptibility of perturbations into account.

- We propose a diverse set of metrics to effectively capture aspects like semantic distance to original, smoothness and grammaticality of perturbed inputs. This is key to understand the perceptibility of small adversarial input perturbations in text.

- We introduce a novel and powerful attack algorithm, CONTEXT-AWAREEXPLANATIONATTACK (CEA), which is shown to consistently outperform state-of-the-art adversaries and therefore allows us to more accurately estimate attribution robustness in text classifiers.

- We are the first to utilize masked language models (MLMs) for context-aware candidate extraction in attribution robustness estimation. This is important because domain-specific MLMs are becoming increasingly available, making them a progressively attractive alternative to less effective, custom synonym embeddings on which current estimation methods have to rely.

- We successfully speed up robustness estimation with the usage of distilled language models and batch masking.

## 2 Related work

The robustness aspect of faithful explanations (Jacovi & Goldberg, 2020) has recently been studied with increasing interest. The authors Ghorbani et al. (2019) were the first to show that attribution methods like Integrated Gradients (Sundararajan et al., 2017) and DeepLIFT (Shrikumar et al., 2017), amongst others, lack robustness to local, imperceptible perturbations in the input that lead to significantly altered attribution maps while maintaining the correct prediction of the image classifier. The works of Dombrowski et al. (2019), Chen et al. (2019), Moosavi-Dezfooli et al. (2019), Rigotti et al. (2022) and Ivankay et al. (2021) have further studied this phenomenon and established theoretical frameworks to understand and mitigate the lack of attribution robustness in the image domain.

However, explanation robustness in natural language processing has not been explored as deeply. The authors Jain & Wallace (2019) and Wiegreffe & Pinter (2020) show that similar inputs can lead to similar attention values but different predictions, and that models can be retrained to yield different attention values for identical inputs and outputs. This, however, does not directly contradict the prediction assumption of faithfulness (Jacovi & Goldberg, 2020) as discussed by Wiegreffe & Pinter (2020). Closer to our work, the works of Ivankay et al. (2022) and Sinha et al. (2021) are the first to prove that explanations in text classifiers are also susceptible to input changes in a very small local neighbourhood of the input. Ivankay et al. (2022) introduce TEXTEXPLANATIONFOOLER (TEF) as a baseline to alter attributions and estimate local robustness of attributions in text. However, the authors' definition of AR does not take semantic distances between original and adversarial samples into account. Moreover, it draws token substitution candidates from a separately trained custom synonym embedding. Thus, their attack results in out-of-context and non-fluent adversarial samples, rendering such perturbations easily detectable. Our work aims to improve the imperceptibility of input alterations and estimate AR with less detectable adversarial alterations that change attributions to a greater extent.

## 3 Preliminaries

A text dataset $\mathbb{S}$ is comprised of $N$ text samples $\boldsymbol{s}_i$, each containing a series of tokens $w_i$ from a vocabulary $\mathbb{W}$ and labels $l_i$ drawn from the label set $\mathbb{L}$. A text classifier $F$ is a function that maps each sample $\boldsymbol{s}_i$ to a label $y_i \in \mathbb{L}$. It consists of an embedding function $E$ and a classifier function $f$. The embedding function $E : \mathbb{S} \to \mathbb{R}^{d \times p}$, $E(\boldsymbol{s}) = \boldsymbol{X}$ maps the text samples $\boldsymbol{s}_i$ to a continuous embedding $\boldsymbol{X}_i$, while the classifier function $f : \mathbb{R}^{d \times p} \to \mathbb{R}^{|\mathbb{L}|}$, $f(\boldsymbol{X}) = \boldsymbol{o}$ maps the embeddings to the output probabilities for each class.

An *attribution function* $A(\boldsymbol{s}, F, l) = \boldsymbol{a}$ assigns a real number to each token $w_j$ in sample $\boldsymbol{s}$. This represents the tokens influence towards the classification outcome. A positive value represents a token that is deemed relevant *towards* the label $l$, a negative value *against* it. We consider the attribution methods Saliency (S) (Simonyan et al., 2013), Integrated Gradients (IG) (Sundararajan et al., 2017) and Self-Attention (A) (Bahdanau et al., 2015).

The *perplexity* (Brown et al.) of a text sample $\boldsymbol{s}$ with tokens $w_j$, given a language model $L$, measures how well the probability distribution given by $L$ predicts the sample $\boldsymbol{s}$, as defined in Equation (1):

$$PP(\boldsymbol{s}|L) = 2^{-\sum_{w_j \in \boldsymbol{s}} p(w_j|L,\boldsymbol{s}) \log p(w_j|L,\boldsymbol{s})} \qquad (1)$$

where $PP$ denotes the perplexity of the text sample $\boldsymbol{s}$ and $p(w_j|L, \boldsymbol{s})$ the probability of token $w_j$ given $L$ and $\boldsymbol{s}$. Low perplexity values indicate that the model $L$ has captured the true distribution of the text dataset $\mathbb{S}$ well.

*Sentence encoders* are embedding functions $E_s : \mathbb{S} \to \mathbb{R}^m$, $E_s(\boldsymbol{s}) = \boldsymbol{e}$ that assign a continuous embedding vector of dimension $m$ to each text sample (Reimers & Gurevych, 2019). These embeddings are used to capture higher-level representations of sentences or short paragraphs that can be used to train downstream tasks effectively. As they are jointly trained on a diverse set of multi-task problems, they are argued to capture the semantic meaning of the text well Reimers & Gurevych (2019).

## 4 Attribution Robustness

In this section, we introduce our novel definition of attribution robustness (AR) in text classifiers. We describe our attribution and text distance measures, which are taken from current work. Furthermore, we describe the optimization problem of estimating AR, our threat model as well as our new estimator algorithm.

### 4.1 Attribution Robustness in Text

Most related works define AR as the maximal attribution distance with a given locality constraint in the search space (Ivankay et al., 2022; Sinha et al., 2021). We argue that this is potentially problematic, as the extent of the input perturbation is not taken into account. Two adversarial samples with similarly altered attributions might in fact strongly differ in terms of how well they maintain semantic similarity to the original sample (see e.g. $3^{\text{rd}}$ example in Figure 1). This suggests that a proper measure of attribution robustness should ascribe higher robustness to methods that are only vulnerable to larger perturbations while being impervious to imperceptible ones. Thus, we give a novel definition for attribution robustness for a given text sample $\boldsymbol{s}$ with true and predicted label $l$ as functions of both resulting attribution distance and input perturbation size, written in Equation (2).

$$r(\boldsymbol{s}) = \max_{\tilde{\boldsymbol{s}} \in \mathcal{N}(\boldsymbol{s})} \frac{d\big[A(\tilde{\boldsymbol{s}}, F, l),\ A(\boldsymbol{s}, F, l)\big]}{d_s(\tilde{\boldsymbol{s}}, \boldsymbol{s})} \tag{2}$$

with the constraint that the predicted classes of $\tilde{\boldsymbol{s}}$ and $\boldsymbol{s}$ are equal, written in Equation (3).

$$\underset{i \in \{1...|\mathbb{L}|\}}{\arg\max} F_i(\tilde{\boldsymbol{s}}) = \underset{i \in \{1...|\mathbb{L}|\}}{\arg\max} F_i(\boldsymbol{s}) \tag{3}$$

Here, $d$ denotes the distance between attribution maps $A(\tilde{\boldsymbol{s}}, F, l)$ and $A(\boldsymbol{s}, F, l)$, $F$ the text classifier with output probability $F_i$ for class $i$, and $d_s$ the distance of input text samples $\tilde{\boldsymbol{s}}$ and $\boldsymbol{s}$. $\mathcal{N}(\boldsymbol{s})$ indicates a neighbourhood of $\boldsymbol{s}$: $\{\mathcal{N}(\boldsymbol{s}) = \tilde{\boldsymbol{s}} \mid d_s(\tilde{\boldsymbol{s}}, \boldsymbol{s}) < \varepsilon\}$ for a small $\varepsilon$. This definition is inspired by the robustness assumption of faithful explanations (Jacovi & Goldberg, 2020). The estimated robustness of an attribution method $A$ on a model $F$ then becomes the expected per-sample $r(\boldsymbol{s})$ on dataset $\mathbb{S}$, see Equation (4).

$$r(A, F) = \mathbb{E}_{\boldsymbol{s} \in \mathbb{S}}\big[r(\boldsymbol{s})\big] \tag{4}$$

We call this $r$ the estimated *attribution robustness (AR) constant*. The robustness of attribution method $A$ on the model $F$ is *inversely proportional* to $r(A, F)$, as high values mean large attribution distances and small input perturbations, which indicates low robustness.

### 4.2 Distances in Text Data

In order to compute the attribution robustness constant $r$ from Equation (4), the distance measures in the numerator and denominator of Equation (2) need to be defined. In explainable AI, it is often argued that only the relative rank between input features or tokens is important when explaining the outcome of a classifier, or even only the top few features. Users frequently focus on the features deemed most important to explain a decision and disregard the less important ones (Ghorbani et al., 2019; Ivankay et al., 2021; Dombrowski et al., 2019). Therefore, it is common practice (Sinha et al., 2021; Ivankay et al.,

2022) to use correlation coefficients and top-k intersections as distance measures between attributions. For this reason, we utilize the Pearson correlation coefficient (PCC) (Pearson, 1895) as attribution distance $d\big[A(\tilde{\boldsymbol{s}}, F, l),\ A(\boldsymbol{s}, F, l)\big] = 1 - \dfrac{1 + \mathrm{PCC}\big[A(\tilde{\boldsymbol{s}}, F, l),\ A(\boldsymbol{s}, F, l)\big]}{2}$ of Equation (2).

The denominator in Equation (2) contains the distance between original and adversarial text samples. In textual input domains, measuring distance between inputs in the adversarial setting is not as straightforward as in the image domain, where $\ell_p$-norm induced distances are common. String distance metrics (Navarro, 2001) can only be used limitedly, as two words can have similar characters but entirely different semantics. For this reason, we propose the following set of measures to effectively capture smoothness, semantic distance to original, and correctness of grammar of adversarial text inputs.

First, we utilize pretrained sentence encoders to measure the semantic textual similarity between the original and adversarial text samples. This can be computed by the cosine similarity between the sentence embeddings of the two text samples, given as

$$d_{\mathrm{s}}(\tilde{\boldsymbol{s}}, \boldsymbol{s}) = 1 - \frac{s_{cos}[E_s(\tilde{\boldsymbol{s}}), E_s(\boldsymbol{s})] + 1}{2} \tag{5}$$

where $d_{\mathrm{s}}$ denotes the semantic distance between samples $\tilde{\boldsymbol{s}}$ and $\boldsymbol{s}$, $s_{cos}$ the cosine similarity, and $E_s(\tilde{\boldsymbol{s}})$ and $E_s(\boldsymbol{s})$ the sentence embeddings of the two input samples. The semantic textual similarity provides a measure how close the two inputs are in their semantic meaning. To this end, the Universal Sentence Encoder (Cer et al., 2018) is widely-used in adversarial text setups (Sun et al., 2020; Ivankay et al., 2022). However, this architecture is not state-of-the-art on the STSBenchmark dataset (Cer et al., 2017), a benchmark used to evaluate semantic textual similarity. Therefore, we utilize a second sentence encoder architecture trained by the authors Wang et al. (2020), MiniLM. This model achieves close to state-of-the-art performance on the benchmark while maintaining a low computational cost.

Our second input distance is derived from the perplexity of original and adversarial inputs $\tilde{\boldsymbol{s}}$ and $\boldsymbol{s}$. We capture the relative increase of perplexity when perturbing the original sentence $\boldsymbol{s}$, given the pretrained GPT-2 language model (Radford et al., 2019) (Equation 6).

$$d_s(\tilde{\boldsymbol{s}}, \boldsymbol{s}) = \frac{PP(\tilde{\boldsymbol{s}}|L) - PP(\boldsymbol{s}|L)}{PP(\boldsymbol{s}|L) + \varepsilon} \tag{6}$$

where $d_{\mathrm{s}}$ denotes the distance between inputs $\tilde{\boldsymbol{s}}$ and $\boldsymbol{s}$, $PP$ the perplexity of the text sample given the GPT-2 language model $L$ and $\varepsilon$ is a small constant. Intuitively, this measure indicates how natural the resulting adversarial inputs are.

Lastly, we capture the increase of grammatical errors in the input samples using the LanguageTool API [1]. As grammatical errors are easily perceived by the human observer, they significantly contribute to the perceptibility of adversarial perturbations (Ebrahimi et al., 2018).

## 4.3 Context-Aware Robustness Estimation

Given our AR definition in Equation (2), in order to estimate the true robustness of an attribution method for a given model, all possible input sequences $\tilde{\boldsymbol{s}}$ within the neighborhood $\mathcal{N}$ of $\boldsymbol{s}$ would have to be checked, which is intractable. Therefore, we restrict the search space to sequences $\tilde{\boldsymbol{s}}$ that only contain token substitutions from the predefined vocabulary set $\mathbb{W}$. Moreover, we restrict the ratio of substituted tokens in the original sequence to $\rho_{max}$, considering only $|\mathbb{C}|$ number of possible substitutions for each token in $\boldsymbol{s}$. The number $|\mathbb{C}|$ is chosen to yield high attribution distance while keeping the computation cost low, detailed in Section 5. This way, we reduce the total perturbation set from $|\mathbb{W}|^{|\boldsymbol{s}|}$ to $|\mathbb{C}|^{|\boldsymbol{s}| \cdot \rho_{max}}$ samples. These are widely used simplifications of the adversarial search in text (Li et al., 2020). The adversarial sequence $\boldsymbol{s}_{\mathrm{adv}}$ then becomes the perturbed sequence that maximizes $r(\boldsymbol{s})$ from Equation (2)

We estimate AR with our novel CONTEXT-AWAREEXPLANATIONATTACK (CEA). CEA is a black-box attack, only having access to the model's prediction and the accompanying attributions, no intermediate representations or gradients. CEA consists of the following two steps.

---

[1] https://languagetool.org

---

**Algorithm 1** `Context-AwareExplanationAttack`

---

**Input**: Input sentence $\boldsymbol{s}$ with label $l$, classifier $F$, attribution $A$, attribution distance $d$, DistilBERT-MLM $L$, number of candidates $N$, maximum perturbation ratio $\rho_{max}$, batch masking ratio $\rho_b$

**Output**: Adversarial sentence $\boldsymbol{s}_{\text{adv}}$

1: $\boldsymbol{s}_{\text{adv}} \leftarrow \boldsymbol{s}$, $d_{max} \leftarrow 0$, $n \leftarrow 0$
2: **for** $w_i \in \boldsymbol{s}$ **do**
3:    $I_{w_i} = d\big[A(\boldsymbol{s}_{w_i \rightarrow 0}, F, l),\ A(\boldsymbol{s}, F, l)\big]$          $\triangleright$ Importance Ranking
4: $\boldsymbol{s}_B \leftarrow \langle \boldsymbol{s}_{1...b}, \boldsymbol{s}_{b+1...2b}, ..., \boldsymbol{s}_{|\boldsymbol{s}|-b+1...|\boldsymbol{s}|} \rangle$ with $I_{w_{b-1}} \geq I_{w_b} \ \forall j \in \{2, ..., |\boldsymbol{s}_B|\}$ and $\forall b \in \{1, .., |\boldsymbol{s}_j|\}$
5: **for** $\boldsymbol{s}_b \in \boldsymbol{s}_B$ **do**
6:    $\mathbb{C}_{\mathbf{b}} \leftarrow L(\boldsymbol{s}_{b \rightarrow [MASK]}, \boldsymbol{s}_{\text{adv}})$         $\triangleright$ Batch Masking and Candidate Extraction
7:    **for** $w_j \in \boldsymbol{s}_b$ **do**
8:       **if** $w_j \in \mathbb{S}_{\texttt{Stopwords}}$ **then**         $\triangleright$ Stop Word Filter
9:          **continue**
10:       **for** $c_k \in \mathbb{C}_j$ **do**         $\triangleright$ Iterate over Candidates
11:          $\tilde{\boldsymbol{s}}_{w_j \rightarrow c_k} \leftarrow$ Replace $w_j$ in $\boldsymbol{s}_{\text{adv}}$ with $c_k$
12:          **if** $\arg\max_{i \in \{1:|\mathbb{L}|\}} F(\tilde{\boldsymbol{s}}_{w_j \rightarrow c_k}) \neq l$ **then**       $\triangleright$ Prediction Filter
13:            **continue**
14:          $\tilde{d} = d\big[A(\tilde{\boldsymbol{s}}_{w_i \rightarrow c_k}, F, l), A(\boldsymbol{s}, F, l)\big]$
15:          **if** $\tilde{d} > d_{max}$ **then**         $\triangleright$ Candidate Selection
16:            $\boldsymbol{s}_{\text{adv}} \leftarrow \tilde{\boldsymbol{s}}_{w_i \rightarrow c_k}$
17:            $d_{max} \leftarrow \tilde{d}$
18:            $n \leftarrow n + 1$
19:       **if** $\rho = \frac{n+1}{|\boldsymbol{s}|} > \rho_{max}$ **then**         $\triangleright$ Limit of Word Substitutions
20:          **break**

---

**Step 1: Word importance ranking.** The first step extracts a priority ranking of tokens in the input text sample $\boldsymbol{s}$. For each word $w_i$ in $\boldsymbol{s}$, CEA computes $I_{w_i} = d\big[A(\boldsymbol{s}_{w_i \rightarrow 0}, F, l),\ A(\boldsymbol{s}, F, l)\big]$, where $\boldsymbol{s}_{w_i \rightarrow 0}$ denotes the token $w_i$ in $\boldsymbol{s}$ set to the zero embedding vector and $d$ denotes the attribution distance measure in Equation (2), described in the previous subsection. The tokens in $\boldsymbol{s}$ are then sorted by descending values of $I_{w_i}$. Thus, we estimate words that are *likely* to result in large attribution distances and prioritize those for substitutions towards building explanation attacks. Importance ranking has been shown to be effective in prioritizing words that yield large changes in the outcomes (Li et al., 2020; Ivankay et al., 2022).

**Step 2: Candidate selection and substitution.** The second step substitutes each highest ranked token in $\boldsymbol{s}$, computed in **Step 1**, with a token from a candidate set $\mathbb{C}$, in descending importance order. The candidate set for a specific word is extracted by first substituting the specific word with the "<MASK>" token, then propagating the whole sentence (with the "<MASK>" token) through a transformer-based masked language model (MLM). The MLM then predicts what tokens or words are the *most likely* to fill in the masked word by assigning a probability distribution over all possible tokens in the vocabulary. CEA takes the $|\mathbb{C}|$ number of tokens with highest probabilities as candidate set to replace the specific word in the sentence. Out of this candidate set $\mathbb{C}$, the final substitution is then selected by maximizing the attribution distance. CEA performs this candidate substitution with the MLM for each highest ranked word in the sentence iteratively in a sequential order. In order to keep the computational costs low, we utilize the DistilBERT pretrained masked language model (Sanh et al., 2019), a BERT-MLM with significantly fewer parameters and more computationally efficient. Also, at most $n = \lfloor |\boldsymbol{s}| \cdot \rho_{max} \rfloor$ words are substituted. While candidate extraction with masked language models has been introduced before, we are the first to apply this concept to the AR estimation problem.

In order to further reduce computational cost, CEA uses batch masking. Thus, instead of masking each word separately in Step 2, the first $n_b = |\boldsymbol{s}| \cdot \rho_b$ most important tokens are masked at once and the language model is queried for candidates for all of these masked tokens. Here, $n_b$ denotes the number, $\rho_b$ the ratio of tokens in $\boldsymbol{s}$ to be masked at once. For instance, during AR estimation of a 100 word text sample, given

$\rho_{max} = 0.15$ and $\rho_b = 0.05$, the MLM is queried only $(100 \cdot 0.15)/(100 \cdot 0.05) = 3$ times with batch masking instead of $100 \cdot 0.15 = 15$ times without it. We compared the runtime of CEA using non-distilled (Devlin et al., 2019) and distilled (Sanh et al., 2019) BERT MLMs, with and without batch masking, and found considerable performance increase with batch masking and distillation. The results are reported in Section 5.

## 5 Experiments

In this section, we present our AR estimation experiments. Specifically, we describe the evaluation setup and results with our novel robustness definition. We show that CEA consistently outperforms our direct state-of-the-art competitor, TEXTEXPLANATIONFOOLER (TEF) in terms of the attribution robustness constant $r$ described in Section 4. Thus, we convey that CEA extracts smoother adversarial samples that are able to alter attributions more significantly than TEF. Finally, we compare the runtime of CEA to TEF and show that CEA achieves comparable runtimes, while still outperforming TEF in the previously mentioned aspects.

### 5.1 Setup

We evaluate the robustness constant $r$ estimated by CEA on the AG's News (Zhang et al., 2015), MR Movie Reviews (Zhang et al., 2015), IMDB (Maas et al., 2011), Yelp (Asghar, 2016) and the Fake News datasets Lifferth (2018). We train a CNN, an LSTM, an LSTM with an attention layer (LSTMAtt), a finetuned BERT (Devlin et al., 2019), RoBERTa (Liu et al., 2019) and XLNet (Yang et al., 2019) classifier for each dataset. A description of these can be found in the appendix. We estimate the robustness of the Saliency (S), Integrated Gradients (IG) and Self-Attention (A) attribution methods. The CNN and LSTM architectures are used in combination with S and IG, the remaining LSTMAtt, BERT, RoBERTA and XLNet are used with all three attributions. Thus, we evaluate 16 combinations of models and attributions for each dataset.

We vary the $\rho_{max}$ parameter of CEA between 0.01 and 0.4. A value of $\rho_{max}$ does not necessarily lead to the actual perturbed ratio of tokens $\rho$ to be $\rho = \rho_{max}$ due to the prediction constraint. We set the batch masking size $\rho_b = \min(\rho_{max}, 0.15)$, as the MLM was trained by masking $\sim 15\%$ of the tokens (Sanh et al., 2019). We set $|\mathbb{C}| = 15$, as larger values do not result in better estimation in terms of $r$, but in significantly higher attack runtimes. This makes our experiments comparable to TEF (Ivankay et al., 2022).

Our attack and experiments are implemented in PyTorch (Paszke et al., 2019), utilizing the Hugging Face Transformer library (Wolf et al., 2020), Captum (Kokhlikyan et al., 2020) and SpaCy (Honnibal et al., 2020). We run each experiment on an NVIDIA A100 GPU with three different seeds and report the average results.

### 5.2 Results

We report the following metrics as functions of the true perturbed ratio $\rho$. The average PCC values of original and adversarial attribution maps indicate the amount of change in explanations. Lower values mean larger attribution changes, thus less robust attribution methods for the given dataset and classification model. The input distance between text samples is captured by the semantic textual similarity values of the original and adversarial samples, measured by the cosine similarity between the USE (Cer et al., 2018) and MiniLM (Wang et al., 2020) sentence embeddings ($SemS_{USE}$ and $SemS_{MiniLM}$), as well as the relative perplexity increase ($\Delta_{PP}$). The average increase in number of grammatical errors ($GE$) after perturbation is also reported. At constant attribution change, higher semantic similarities and lower perplexities indicate lower attribution robustness, as *smaller, more imperceptible alterations* are enough to change the outcome of the attributions.

Using the aforementioned values, we report the estimated robustness constants $r_{USE}$, $r_{MiniLM}$ and $r_{PP}$, according to Equation (4). We compare these metrics for our novel CEA algorithm and the direct competitor TEF (Ivankay et al., 2022). The results are reported in Figure 2. The continuous lines contain the metrics for our CEA attack, the dashed lines for the baseline TEF. The figures show that CEA perturbations alter explanations more (lower PCC values) while yielding adversarial samples semantically equally or more similar to the original inputs than TEF (higher average $SemS$, lower average $\Delta_{PP}$ and $GE$ values). Moreover, the

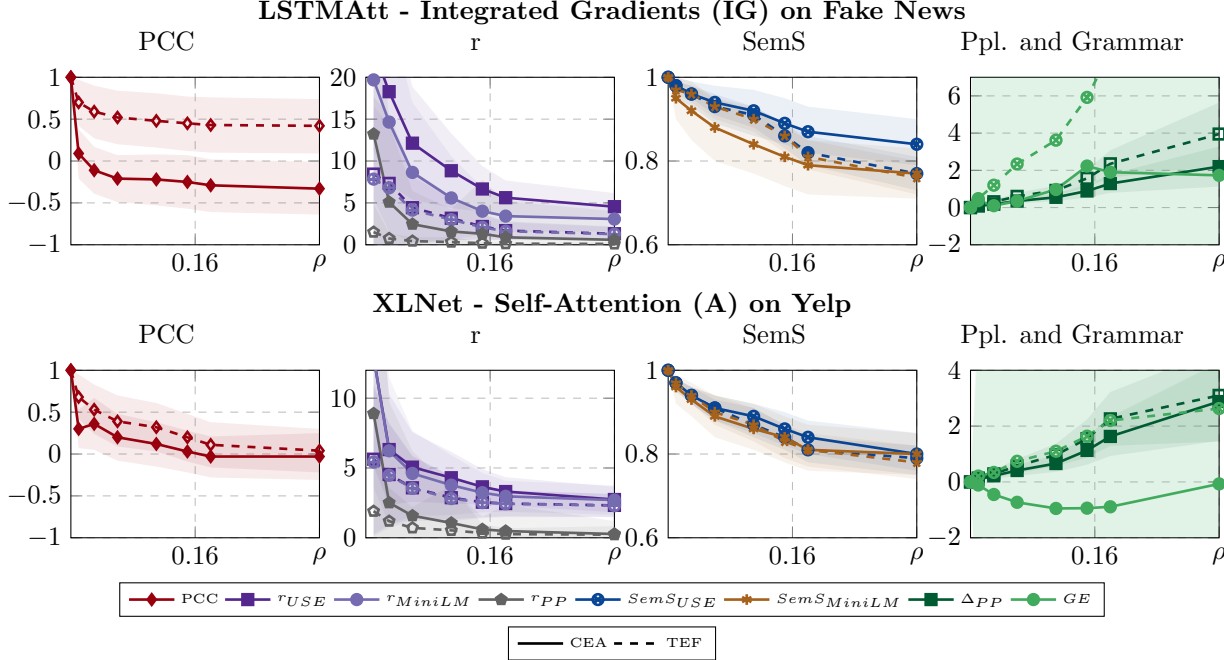

Figure 2: AR metrics as functions of the ratio of perturbed tokens $\rho$. We plot the mean and standard deviation of the Pearson correlations (PCC) between original and adversarial attributions, the estimated AR robustness constants ($r$), the semantic similarities ($SemS$), relative perplexity increase ($\Delta_{PP}$) and increase of number of grammatical errors ($GE$) in original and adversarial text inputs. We compare these values for our novel CONTEXT-AWAREEXPLANATIONATTACK (CEA - continuous lines) and the baseline TEXTEXPLA-NATIONFOOLER (TEF - dashed lines). We observe consistent improvement in robustness estimation with CEA compared to TEF, reflected in higher $r$-values in the second column. This is attributed to both lower PCC values, higher semantic similarities of perturbed sentences to the original ones and lower adversarial perplexity of CEA perturbations.

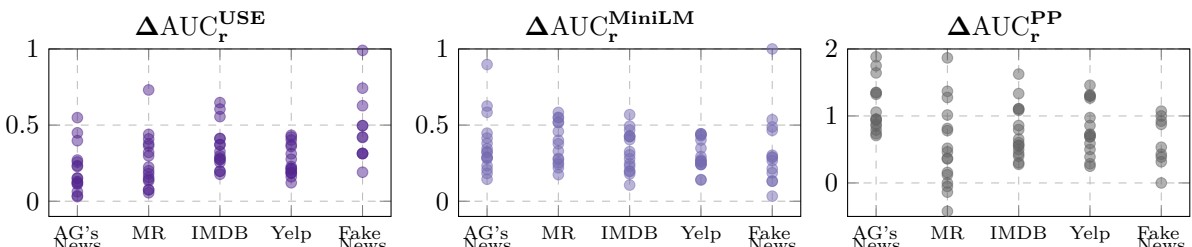

Figure 3: Relative increase $\Delta$ of $AUC_r$ when estimating the robustness constants $r$ (Equation 4) with CEA compared to TEF. Each point corresponds to one of the 16 combinations of model and attribution method, on the indicated dataset. The $r$-values are estimated with the PCC as attribution similarity, varying the input distance measures $d_s$ as described in Section 4.2. We observe a relative increase of $0.3 - 1.5$ for almost all models, attribution maps and datasets evaluated on. This shows that CEA consistently provides better perturbations that alter attributions more while being more fluent and semantically similar to the unperturbed input.

perplexity increase is consistently lower for CEA perturbations, leading to more fluent adversarial samples. This is well-captured by resulting robustness constants $r$, which are higher for CEA than TEF, showing both that our AR definition of Equation (2) is a suitable indicator for AR in text classifiers, and that CEA estimates this robustness better than the state-of-the-art TEF attack. The rest of the results is reported in the appendix.

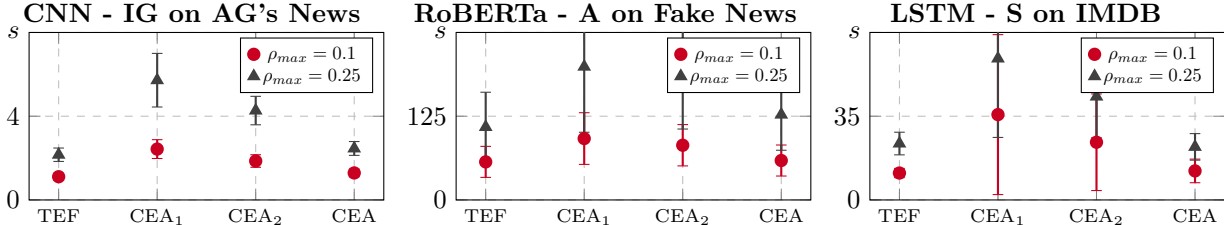

Figure 4: Per-sample runtime (s) of our AR estimator algorithm versions. CEA, with a distilled MLM and batch masking, achieves comparably fast estimation to TEF, while CEA with a non-distilled BERT MLM ($CEA_1$) is the slowest estimator, with a relative increase in runtime of approx. 1.5-2.5 compared to TEF. Distillation of the MLM ($CEA_2$) improves the runtime by around 25-35% compared to ($CEA_1$).

### 5.2.1 Area Under the Curves

To quantify the overall performance of CEA over the whole operation interval of $\rho$, we compute the area under the estimated $r$ curves ($2^{nd}$ column in Figure 2). These are calculated as the integral $AUC_r = \int_\rho r(A, F)d\rho$. High $AUC_r$ values correspond to high $r$-values, thus low overall attribution robustness. We then compare the resulting $AUC_r$ estimated with our CEA algorithm to the competitor method TEF. Figure 3 shows the relative increase of AUC when estimating with CEA rather than TEF, for each of the 16 combinations of models and attribution methods for a given dataset. For instance, a value of 0.5 indicates an increase of 50% in estimated $AUC_r$, i.e. if TEF results in $AUC_r = 1.0$, CEA yields $AUC_r = 1.5$. We plot the $AUC_r$ increase estimated with the semantic textual similarities from USE ($AUC_r^{\mathbf{USE}}$), MiniLM ($AUC_r^{\mathbf{MiniLM}}$) and with the relative perplexity increase ($AUC_r^{\mathbf{PP}}$). The attribution distance in the numerator of $r$ is set to the PCC, described in Section 4. We observe an increase in $AUC_r$ of $0.3 - 0.5$ with USE and MiniLM, and $0.5 - 1.5$ with PP for most models, attribution maps and datasets. This further shows that CEA consistently yields higher robustness constants $r$ than TEF, providing better perturbations that alter attributions more while being less perceptible.

### 5.2.2 Runtime Analysis

Querying transformer-based MLMs is computationally expensive. Substituting the synonym extraction from TEF with an MLM-based candidate extraction results in a significant increase in estimation time. Therefore, we use the methods described in Section 4 to lower the estimation time in CEA. Figure 4 contains the per-sample attack time for TEF, CEA with the non-distilled BERT MLM ($CEA_1$), CEA with DistilBERT MLM ($CEA_2$) and our CEA algorithm with DistilBERT MLM and batch masking, for $\rho_{max} \in \{0.1, 0.25\}$. We observe that $CEA_1$ results in a significant increase in mean estimation time by a factor of around 2 compared to TEF on both a smaller, medium and a large datasets. Using $CEA_2$ for estimating AR decreases the runtime by a large margin compared to $CEA_1$. Finally, when applying both a distilled MLM and batch masking - CEA, the per-sample attack time is comparable to the baseline TEF, while maintaining better AR estimation.

### 5.2.3 Ablation Studies

CEA differs from our direct competitor TEF (Ivankay et al., 2022) in Step 2 of the algorithms. Instead of utilizing the synonym embeddings Mrkšic et al. (2016) to extract substitution candidates and passing those through a part of speech filter, CEA uses MLMs to extract the candidates. Thus, our ablations focus around this aspect. We compare TEFs AR performance to two versions of CEA, the original as formulated in Algorithm 1 and one where the candidate extraction is still performed with an MLM as in Algorithm 1, but the selection is random (i.e. Line 14-16). We do not experiment with ablating the stop word filter of the prediction filter, as those are assumptions of robustness and constraints of the optimization problem, not directly design choices of CEA. Figure 5 compares the AR metrics of these three estimators and reports them as functions of $\rho$. We observe that CEA outperforms both TEF and the MLM-based random synonym selection, supporting the choice of MLM-based candidate extraction over TEF's synonym embeddings.

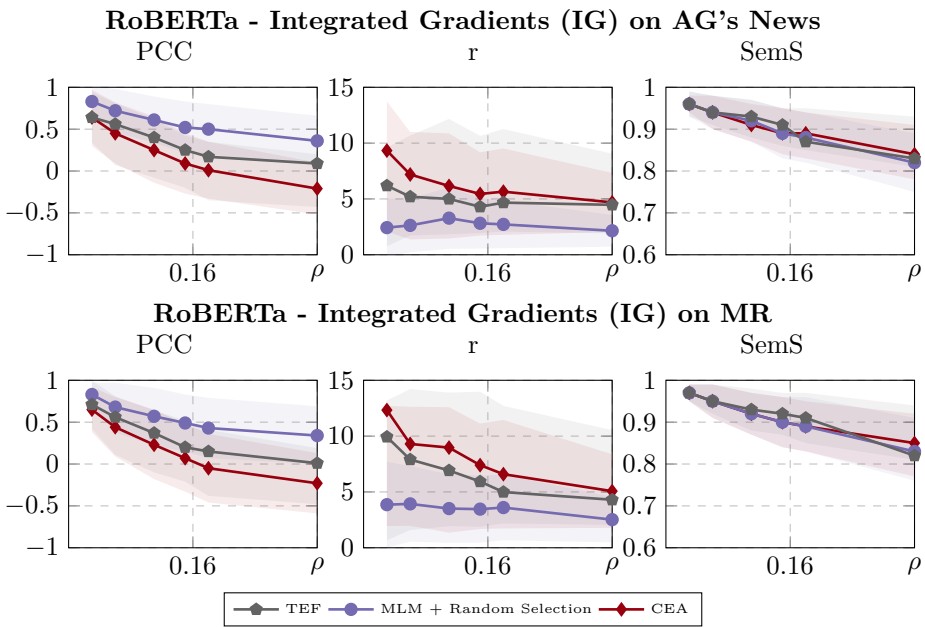

Figure 5: AR estimation performance of CEA, TEF and our ablated CEA with random candidate selection. We observe that CEA out performs TEF both in terms of PCC as well as $r$, indicating the superior performance of MLM-based candidate selection over pretrained, counter-fitted synonym embeddings. However, randomly selecting the substitutions from the candidate set yields worse performance than TEF.

Randomly selecting the substitution from the candidate set significantly speeds up AR estimation, yields however inferior results to both TEF and CEA in terms of both PCC and $r$.

## 6 Conclusion

In this work, we introduced a novel definition of attribution robustness in text classifiers. Crucially, our definition incorporates perturbation size, which contributes significantly to the perceptibility of attacks. We introduce semantic textual similarity measures, the relative perplexity increase and the number of grammatical errors as ways to effectively quantify perturbation size in text. Next, we introduced CONTEXT-AWAREEXPLANATIONATTACK, a new state-of-the-art attack method that results in a tighter estimator for attribution robustness in text classification problems. It is a black-box estimator using a distilled MLM with batch masking to extract adversarial perturbations with small computational overhead. Finally, we showed that our new algorithm CEA outperforms current attacks by altering DNN attributions more with less perceptible perturbations.

One important question arises from the robustness assumption of interpretations: are more robust explanations indeed more faithful? Current work has already started to look into this research question. The authors Ivankay et al. (2023) examine the interplay between robustness and plausibility. However, understanding the impact of robustness on the faithfulness of explanation still remains an open question that we plan to examine in future work.

To sum up, our contributions allow for estimating the robustness of attributions more accurately and are a first step towards training robust, safely applicable DNNs in critical areas like medicine, law or finance.

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

# A  Appendix

## A.1  Study on Randomized Explanations

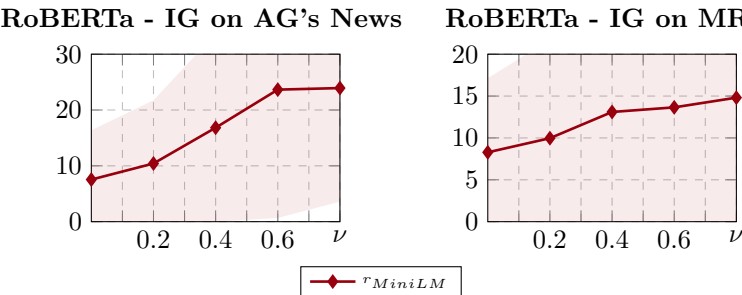

Figure 6: Attribution robustness metric $r$ as function of the ratio of randomized word attributions $\nu$. During AR estimation with CEA, we set a certain ratio of word attributions to a random number in [-1, 1]. A value $\nu = 0.4$ corresponds to 40% of word attributions being random. Our AR metric $r$ positively correlates with $\nu$. This supports our argument that the metric is a suitable measure of AR, as higher $r$ values indicate less robust attributions, which is the case for higher $\nu$-s, given our assumption that *bad quality* explanations are less robust than *good quality* ones.

This experimental study shows how our CEA algorithm behaves in, and our AR metric correlates with, cases where the models fail to give correct explanations. Even though assessing the true *quality* (for instance faithfulness or completeness) of attribution methods used is out of scope for this work, we would like to understand how our metric correlates with partially randomized explanations. We assume that robustness of explanations correlates with their quality, and random attributions do not reflect the true decision process, thus are *bad quality* explanations. Therefore, a metric that represents AR well would correlate with the amount of randomness in the explanations. Higher randomness would indicate lower robustness. In Figure 6, we examine the behaviour of $r$ as a function of $\nu$, the ratio of randomized word attributions in each sentence. We observe a positive correlation between $r$ and $\nu$, which supports our hypothesis that $r$ is a good measure for AR and reflects the correlation between AR and quality of explanations well.

## A.2  Datasets

We estimate the robustness of our attribution methods and models on five publicly available datasets. These are AG's News, MR movie review, IMDB movie review, Yelp and Fake News, all of which are in English. AG's News consists of 127552 news article samples, categorized into the classes World, News, Business and Sci/Tech. We use the concatenation of title and text of the samples to feed into our text classifiers, stripping any sample that is longer than 64 tokens. The MR Movie Review dataset contains 10592 short samples of positive or negative movie reviews. We only use the first 32 tokens in each sample as input to the classifiers. IMDB Movie Review is a dataset consisting of 49952 positive and negative movie reviews, with a maximum token length of 256. Yelp categorizes 700000 reviews of several topics into 5 classes, each representing a rating from 1 to 5. We strip the samples to a maximum length of 256. Fake News is a collection of 20080 news samples, each categorized into reliable or unreliable. These are rather long articles, thus we use a maximum sequence length of 512 for this dataset.

We apply basic preprocessing to all samples in each dataset, which includes converting them to lowercase, removing any special characters not in the English alphabet and emojis. We use 60% of the samples for training the classifier models, 20% for validation and 20% for testing and estimating the robustness of attribution methods.

## A.3  Models

As described in the main paper, we train six classification architectures for each dataset, three DNN-based architectures, which are a CNN, an LSTM, an LSTM with an attention layer (LSTMAtt), as well as three

| Dataset | CNN | LSTM | LSTMAtt | BERT | RoBERTa | XLNet |
|---------|-----|------|---------|------|---------|-------|
| AG's News | 89.7% | 90.8% | 91.4% | 94.2% | 94.0% | 93.8% |
| MR | 73.0% | 76.4% | 78.0% | 82.2% | 87.7% | 86.3% |
| IMDB | 82.0% | 87.2% | 87.3% | 89.4% | 93.3% | 93.7% |
| Yelp | 49.0% | 54.8% | 60.0% | 62.6% | 67.6% | - |
| Fake News | 98.9% | 99.6% | 99.6% | 99.8% | 100.0% | 100.0% |

Table 1: Accuracies of each classifier trained. Our models achieve comparable results to state-of-the-art performance for each dataset.

| | | AG's News | MR | IMDB | Yelp | Fake News |
|---|---|---|---|---|---|---|
| **CNN** | Input shape | (64, 300) | (32,300) | (256, 300) | (256, 300) | (512, 300) |
| | Num. classes | 4 | 2 | 2 | 5 | 2 |
| | Filter sizes | [3, 5, 7] | [3, 5] | [3, 5, 7] | [3, 5, 7] | [3, 5, 7] |
| | Feature sizes | [8, 8, 8] | [8, 8] | [16, 16, 16] | [128, 128, 128] | [32, 32, 32] |
| | Pooling sizes | [2, 2, 2] | [2, 2] | [2, 2, 2] | [2, 2, 2] | [2, 2, 2] |
| | Lin. layer dim. | 8 | 8 | 16 | 64 | 32 |
| | Num. params | 67748 | 27946 | 567458 | 16428293 | 4091714 |
| **LSTM** | Input shape | (64, 300) | (32,300) | (256, 300) | (256, 300) | (512, 300) |
| | Num. classes | 4 | 2 | 2 | 5 | 2 |
| | Hidden dim. | 8 | 8 | 16 | 256 | 16 |
| | Num. layers | 1 | 1 | 2 | 2 | 1 |
| | Pooling sizes | 2 | 2 | 1 | 2 | 2 |
| | Lin. layer dim. | 8 | 8 | 16 | 32 | 16 |
| | Num. params | 10988 | 10458 | 18162 | 2146693 | 85986 |
| **LSTMAtt** | Input shape | (64, 300) | (32,300) | (256, 300) | (256, 300) | (512, 300) |
| | Num. classes | 4 | 2 | 2 | 5 | 2 |
| | Hidden dim. | 8 | 8 | 16 | 256 | 16 |
| | Num. layers | 4 | 1 | 2 | 2 | 1 |
| | Lin. layer dim. | 8 | 8 | 16 | 32 | 16 |
| | Num. params | 25004 | 19994 | 47666 | 2752901 | 41826 |
| **BERT** | Input shape | (64,) | (32,) | (256,) | (256,) | (512,) |
| | Num. classes | 4 | 2 | 2 | 5 | 2 |
| | Model ID | bert-base-uncased | | | | |
| | Num. params | 109485316 | 109483778 | 109483778 | 109486085 | 109483778 |
| **RoBERTa** | Input shape | (64,) | (32,) | (256,) | (256,) | (512,) |
| | Num. classes | 4 | 2 | 2 | 5 | 2 |
| | Model ID | roberta-base | | | | |
| | Num. params | 124648708 | 124647170 | 124647170 | 124649477 | 124647170 |
| **XLNet** | Input shape | (64,) | (32,) | (256,) | (256,) | (512,) |
| | Num. classes | 4 | 2 | 2 | 5 | 2 |
| | Model ID | xlnet-base-cased | | | | |
| | Num. params | 117312004 | 117310466 | 117310466 | 117312773 | 117310466 |

Table 2: Model specifications

transformer-based architectures, which are a finetuned BERT, RoBERTa and XLNet. The CNN, LSTM and LSTMAtt architectures use the 6B-300-dimensional Glove word embeddings, while the transformer-based architectures use the pretrained Hugging Face embeddings of the respective base-uncased versions. The DNN-based classifiers each contain a linear layer on top of their feature extractors and use the built-in SpaCy English tokenizer, the transformers directly map the feature outputs to the output logits with a fully-connected layer and utilize the Hugging Face pretrained tokenizers for each architecture respectively. Table 2 contains the model specifications. We train each model with a standard learning rate of 0.001, using the Adam optimizer with the cross-entropy loss and early stopping. We utilize NVIDIA A100 GPUs to speed up training and AR estimation. The resulting accuracies of the models can be found in Table 1.

## A.4 Additional Examples

| Original sample | CEA perturbed sample (ours) | TEF perturbed sample (Ivankay et al., 2022) |
|---|---|---|
| with the nations media raining heavy criticism down upon him, spain coach luis aragones chose to pin a galling nil-nil uefa world cup qualifying result with lithuania on the large playing

F($s$, "Sports") = 1.00 | with the nations media raining heavy criticism down upon him, spain coach luis aragones chose to pin a galling nil-nil uefa world junior classification result with lithuania on the large yellow

F($s$, "Sports") = 1.00
**r**: 31.69
*SemS*: 0.98
*PCC*: -0.22 | with the nations media raining heavy criticism down upon him, spain buses luis aragones chose to pin a galling nil-nil uefa world goblet qualifying result with lithuania on the large replay

F($s$, "Sports") = 1.00
**r**: 5.37
*SemS*: 0.95
*PCC*: 0.42 |
| when stonehill hired chris woods as its football coach after last season, the hope was he could once again revive a disappointing
F($s$, "Sports") = 1.00 | when Rutgers hired chris woods as its head coach after last season, the hope was he could once again revive a disappointing

F($s$, "Sports") = 1.00
**r**: 3.22
*SemS*: 0.88
*PCC*: 0.25 | when vassar hired chris woods as its balloon coach after last season, the hope was he could once again revive a disappointing

F($s$, "Sports") = 1.00
**r**: 2.96
*SemS*: 0.85
*PCC*: 0.12 |
| the space shuttle will not fly before may 2005, according to nasa officials. this pushes the shuttle #39;s return-to-flight schedule back by two months, and postpones a vital servicing mission to the international space
F($s$, "Sci/Tech") = 1.00 | the Atlantis capsules will not fly before may 2005, according to nasa officials. this pushes the ISS #39;s return-to-flight schedule back by two months, and postpones a vital servicing mission to the international space
F($s$, "Sci/Tech") = 1.00
**r**: 1.46
*SemS*: 0.85
*PCC*: 0.57 | the separation shuttles will not fly before may 2005, according to nasa officials. this pushes the ferry #39;s return-to-flight schedule back by two months, and postpones a vital servicing mission to the international space
F($s$, "Sci/Tech") = 1.00
**r**: 3.62
*SemS*: 0.88
*PCC*: 0.15 |
| dueling cisco systems inc. and juniper networks inc. are both jockeying for the spotlight on the high end of the routing market with announcements of new developments around their respective crs-1 and t-series core
F($s$, "Sci/Tech") = 0.98 | The cisco systems inc. and juniper networks inc. are both jockeying for the spotlight on the high end of the networking market with announcements of new developments around their respective crs-1 and t-series core
F($s$, "Sci/Tech") = 0.99
**r**: 2.90
*SemS*: 0.93
*PCC*: 0.61 | jousting belkin systems inc. and juniper grids inc. are both jockeying for the spotlight on the high end of the routing market with announcements of new developments around their respective crs-1 and t-series core
F($s$, "Sci/Tech") = 0.99
**r**: 4.58
*SemS*: 0.90
*PCC*: 0.04 |
| playboy enterprises inc. (pla.n: quote, profile, research) , the adult entertainment company, on tuesday reported a third-quarter profit, reversing a year-earlier

F($s$, "Business") = 1.00 | playboy enterprises inc. (pla.n: quote, profile, research) , the largest tech company, on tuesday reported a third-quarter profit, reversing a year-earlier

F($s$, "Business") = 0.98
**r**: 2.01
*SemS*: 0.98
*PCC*: 0.92 | playboy enterprises inc. (pla.n: quote, profile, research) , the adulthood entertainment company, on yesterday reported a third-quarter profit, reversing a year-earlier

F($s$, "Business") = 1.00
**r**: 21.20
*SemS*: 0.99
*PCC*: 0.65 |

| Original sample | CEA perturbed sample (ours) | TEF perturbed sample (Ivankay et al., 2022) |
|---|---|---|
| jimmie johnson has fought through mistakes, mechanical failures and the despair of losing friends in a plane crash to charge back into nascar #39;s closest championship battle

$F(s, \text{"Sports"}) = 1.00$ | jimmie johnson has fought through mistakes, mechanical failures and the despair of losing friends in a plane crash to charge back into Halo #39;s closest Halo battle

$F(s, \text{"Sports"}) = 0.56$
**r**: 2.46
*SemS*: 0.90
*PCC*: 0.53 | jimmie johnson has fought through mistakes, mechanical failures and the despair of losing friends in a plane crash to charge back into daytona #39;s closest champion battle
$F(s, \text{"Sports"}) = 1.00$
**r**: 27.69
*SemS*: 0.98
*PCC*: 0.11 |
| islamabad : pakistan and afghanistan have reaffirmed they are partners in fighting terrorism, afghan president hamid karzai declared at the end of a two-day
$F(s, \text{"World"}) = 1.00$ | islamabad : pakistan and afghanistan have reaffirmed they are partners in fighting terrorism afghan ia hamid karzai declared at the end of a two-day
$F(s, \text{"World"}) = 1.00$
**r**: 61.30
*SemS*: 1.00
*PCC*: 0.43 | islamabad : pakistan and afghanistan have reaffirmed they are allies in fighting terrorism, afghan chairmen hamid karzai declared at the end of a two-day
$F(s, \text{"World"}) = 1.00$
**r**: 11.94
*SemS*: 0.97
*PCC*: 0.31 |
| lusty koalas in southern australia are going to be put on the pill to stop them breeding too quickly and putting too much strain on their eucalyptus-forest
$F(s, \text{"Sci/Tech"}) = 1.00$ | lusty koalas in southern australia are going to be put on the spot to stop them disappearing too quickly and putting too much strain on their eucalyptus-forest
$F(s, \text{"Sci/Tech"}) = 0.89$
**r**: 8.67
*SemS*: 0.94
*PCC*: -0.02 | lusty koalas in southern australia are going to be put on the tablet to stop them rearing too quickly and putting too much strain on their eucalyptus-forest
$F(s, \text{"Sci/Tech"}) = 1.00$
**r**: 9.29
*SemS*: 0.94
*PCC*: -0.10 |
| embarcadero technologies on monday is unveiling its dbartisan workbench 8.0 database administration tool, featuring enhanced backup capabilities for microsoft sql server databases and support for performance metrics in the oracle10g
$F(s, \text{"Sci/Tech"}) = 0.99$ | database technologies vendor monday is showcasing its dbartisan workbench 8.0 database administration tool, featuring enhanced backup capabilities for microsoft sql server databases and support for performance metrics in the oracle10g
$F(s, \text{"Sci/Tech"}) = 0.99$
**r**: 4.38
*SemS*: 0.92
*PCC*: 0.33 | alameda techs on monday is brandishing its dbartisan workbench 8.0 database administration tool, featuring enhanced backup capabilities for microsoft sql server databases and support for performance metrics in the oracle10g
$F(s, \text{"Sci/Tech"}) = 0.99$
**r**: 2.08
*SemS*: 0.88
*PCC*: 0.49 |
| in a move likely to have major ramifications for the library world, google announced december 14 that it would embark on an ambitious project to digitally scan books from the collections of five major research libraries and make them searchable
$F(s, \text{"Sci/Tech"}) = 0.94$ | in a move likely to have major repercussions for the digital world, Cambridge announced december 14 that it would embark on an ambitious project to digitally scan data from the collections of five major research libraries and make them searchable
$F(s, \text{"Sci/Tech"}) = 1.00$
**r**: 4.21
*SemS*: 0.91
*PCC*: 0.20 | in a move likely to have major implications for the library world, iphone announced december 14 that it would embark on an ambitious plans to digitally scan livres from the collections of five major research libraries and make them searchable
$F(s, \text{"Sci/Tech"}) = 1.00$
**r**: 3.71
*SemS*: 0.83
*PCC*: -0.25 |

| Original sample | CEA perturbed sample (ours) | TEF perturbed sample (Ivankay et al., 2022) |
|---|---|---|
| stitch is a bad mannered , ugly and destructive little * * * * . no cute factor here . not that i mind ugly ; the problem

F($s$, "Negative") = 1.00 | stitch is a bad mannered , ugly and cute little * * * * . no limiting factor here . not that i mind ugly ; the problem

F($s$, "Negative") = 0.98
**r**: 18.75
*SemS*: 0.99
*PCC*: 0.54 | stitch is a bad mannered , ugly and detrimental little * * * * . no lovely factor here . not that i mind ugly ; the problem

F($s$, "Negative") = 1.00
**r**: 2.21
*SemS*: 0.98
*PCC*: 0.93 |
| miyazaki has created such a vibrant , colorful world , it's almost impossible not to be swept away by the sheer beauty of his images

F($s$, "Positive") = 1.00 | miyazaki has created such a vibrant ly imaginative world , it's almost impossible not to be swept away by the sheer beauty of his images

F($s$, "Positive") = 1.00
**r**: 6.26
*SemS*: 0.97
*PCC*: 0.64 | miyazaki has created such a bustling , picturesque world , it's almost impossible not to be swept away by the sheer beauty of his images

F($s$, "Positive") = 1.00
**r**: 3.67
*SemS*: 0.98
*PCC*: 0.82 |
| it is a challenging film , if not always a narratively cohesive one
F($s$, "Positive") = 1.00 | it is a beautiful film , if not always a narratively cohesive one
F($s$, "Positive") = 1.00
**r**: 2.31
*SemS*: 0.93
*PCC*: 0.68 | it is a problematic film , if not always a narratively cohesive one
F($s$, "Positive") = 0.98
**r**: 2.89
*SemS*: 0.91
*PCC*: 0.50 |
| much like its easily dismissive take on the upscale lifestyle , there isn't much there here
F($s$, "Negative") = 1.00 | much like its easily readable take on the upscale lifestyle , there isn't much there here
F($s$, "Negative") = 1.00
**r**: 4.53
*SemS*: 0.93
*PCC*: 0.40 | much like its easily snide take on the upscale lifestyle , there isn't much there here
F($s$, "Negative") = 1.00
**r**: 4.67
*SemS*: 0.95
*PCC*: 0.53 |
| it's a nicely detailed world of pawns , bishops and kings , of wagers in dingy backrooms or pristine forests
F($s$, "Positive") = 1.00 | it's a surprisingly rich world of pawns , bishops and kings , of wagers in dingy backrooms or pristine forests
F($s$, "Positive") = 1.00
**r**: 8.09
*SemS*: 0.96
*PCC*: 0.35 | it's a politely thorough world of pawns , bishops and kings , of wagers in dingy backrooms or pristine forests
F($s$, "Positive") = 0.98
**r**: 20.51
*SemS*: 0.97
*PCC*: -0.04 |
| an atonal estrogen opera that demonizes feminism while gifting the most sympathetic male of the piece with a nice vomit bath at his wedding
F($s$, "Negative") = 1.00 | an appalling estrogen opera that demonizes feminism while distracting the most sympathetic male of the piece with a nice vomit bath at his wedding
F($s$, "Negative") = 1.00
**r**: 11.64
*SemS*: 0.97
*PCC*: 0.25 | an atonal hormone teatro that demonizes feminism while gifting the most sympathetic male of the piece with a nice vomit bath at his wedding
F($s$, "Negative") = 1.00
**r**: 7.93
*SemS*: 0.96
*PCC*: 0.35 |

| Original sample | CEA perturbed sample (ours) | TEF perturbed sample (Ivankay et al., 2022) |
|---|---|---|
| a brutal and funny work . nicole holofcenter , the insightful writer/director responsible for this illuminating comedy doesn't wrap the proceedings up neatly

F($s$, "Positive") = 1.00 | a brilliant and funny work . nicole holofcenter , the insightful writer/director responsible for this illuminating comedy doesn't wrap the proceedings up ...

F($s$, "Positive") = 1.00
**r**: 14.80
*SemS*: 0.98
*PCC*: 0.52 | a barbaric and funny work . nicole holofcenter , the insightful writer/director responsible for this illuminating comedy doesn't wrap the proceedings up pleasantly

F($s$, "Positive") = 1.00
**r**: 7.92
*SemS*: 0.97
*PCC*: 0.60 |
| leigh isn't breaking new ground , but he knows how a daily grind can kill love

F($s$, "Positive") = 1.00 | leigh isn't breaking new ground , but he knows how a daily workout can kill love

F($s$, "Positive") = 1.00
**r**: 9.09
*SemS*: 0.96
*PCC*: 0.27 | leigh isn't breaking new ground , but he knows how a daily smoothing can kill love

F($s$, "Positive") = 1.00
**r**: 3.94
*SemS*: 0.96
*PCC*: 0.69 |
| new ways of describing badness need to be invented to describe exactly how bad it is

F($s$, "Negative") = 1.00 | new ways of describing cancer need to be invented to describe exactly how bad it is

F($s$, "Negative") = 1.00
**r**: 1.13
*SemS*: 0.82
*PCC*: 0.60 | new ways of describing perversity need to be invented to describe exactly how bad it is

F($s$, "Negative") = 1.00
**r**: 1.45
*SemS*: 0.90
*PCC*: 0.70 |
| to the degree that ivans xtc . works , it's thanks to huston's revelatory performance

F($s$, "Positive") = 1.00 | to the degree that ivans xtc . works , it's thanks to huston's outstanding performance

F($s$, "Positive") = 1.00
**r**: 16.14
*SemS*: 0.98
*PCC*: 0.50 | to the degree that ivans xtc . works , it's thanks to huston's revelatory execution

F($s$, "Positive") = 1.00
**r**: 4.44
*SemS*: 0.96
*PCC*: 0.60 |
| quiet , adult and just about more stately than any contemporary movie this year . . . a true study , a film with a questioning heart and

F($s$, "Positive") = 1.00 | mature , funny and just about more stately than any contemporary movie this year . . . a true study , a film with a questioning heart and

F($s$, "Positive") = 1.00
**r**: 1.83
*SemS*: 0.95
*PCC*: 0.82 | quiet , adulthood and just about more stately than any topical movie this year . . . a true study , a film with a questioning heart and

F($s$, "Positive") = 1.00
**r**: 1.37
*SemS*: 0.97
*PCC*: 0.92 |
| a markedly inactive film , city is conversational bordering on confessional


F($s$, "Negative") = 1.00 | a markedly inactive neighbourhood , city is conversational bordering on confessional

F($s$, "Negative") = 1.00
**r**: 3.92
*SemS*: 0.91
*PCC*: 0.32 | a markedly idle film , city is conversational bordering on confessional


F($s$, "Negative") = 1.00
**r**: 17.79
*SemS*: 0.98
*PCC*: 0.21 |
| an entertaining , if somewhat standardized , action movie

F($s$, "Positive") = 1.00 | an excellent , if somewhat standardized , action movie

F($s$, "Positive") = 1.00
**r**: 3.65
*SemS*: 0.94
*PCC*: 0.59 | an entertain , if somewhat standardized , action movie

F($s$, "Positive") = 0.99
**r**: 11.43
*SemS*: 0.97
*PCC*: 0.22 |

| Original sample | CEA perturbed sample (ours) | TEF perturbed sample (Ivankay et al., 2022) |
|---|---|---|
| wonder of wonders – a teen movie with a humanistic message

F($s$, "Positive") = 1.00 | Land of wonders – a teen movie with a humanistic message

F($s$, "Positive") = 1.00
**r**: 9.92
*SemS*: 0.97
*PCC*: 0.32 | astonishment of wonders – a teen movie with a humanistic message

F($s$, "Positive") = 1.00
**r**: 7.05
*SemS*: 0.94
*PCC*: 0.21 |
| star trek was kind of terrific once , but now it is a copy of a copy of a copy

F($s$, "Negative") = 1.00 | star trek was kind of lame once , but now it is a hell of a copy of a copy

F($s$, "Negative") = 0.81
**r**: 20.64
*SemS*: 0.96
*PCC*: -0.57 | star trek was kind of superb once , but now it is a copies of a copy of a copy

F($s$, "Negative") = 1.00
**r**: 4.23
*SemS*: 0.98
*PCC*: 0.82 |

## A.5   Additional AR Results

As described in the main body of our paper, we plot the Pearson Correlation Coefficient between original and adversarial attribution values of the words (1st column from left), the estimated robustness constants $r$ (2nd column from left) as well as the semantic similarities between unperturbed and perturbed input texts, the perplexity increase and the increase in number of grammatical errors (3rd and 4th column from left) after perturbation. We consider a high estimated robustness constant $r$ as *successful* attack, thus low PCC values accompanied by high semantic similarities, low perplexity increase values and grammatical errors. Based on the graph below, we conclude that CEA consistently yields higher estimated robustness constants $r$ than the reference method TEF, due to lower Pearson correlation between adversarial and original attribution maps, higher semantic similarities and smaller perplexity increases after applying the adversarial perturbations.

### A.5.1   AG's News

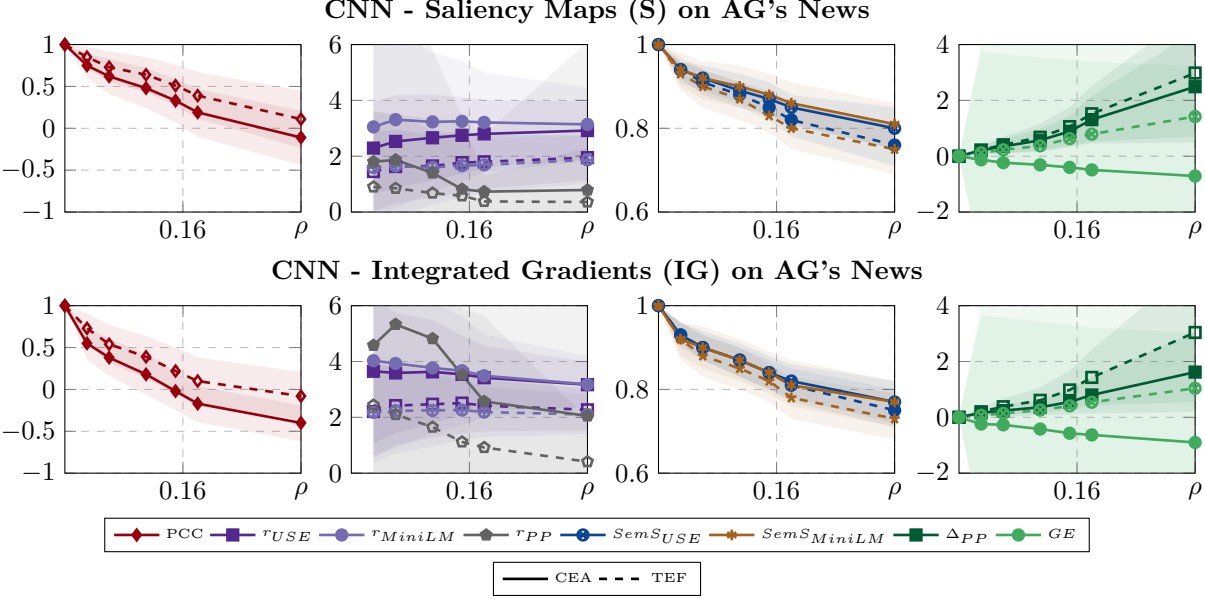

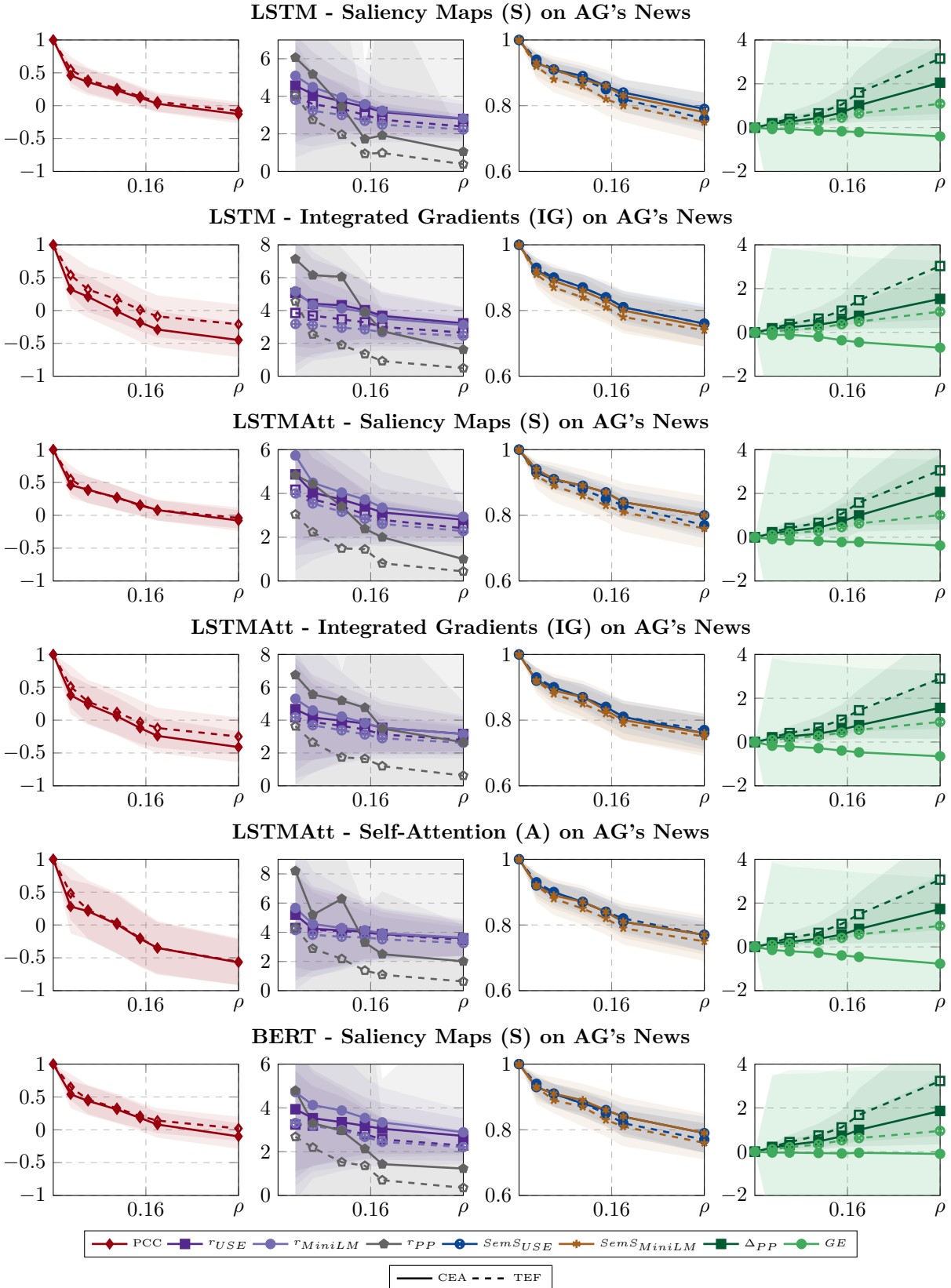

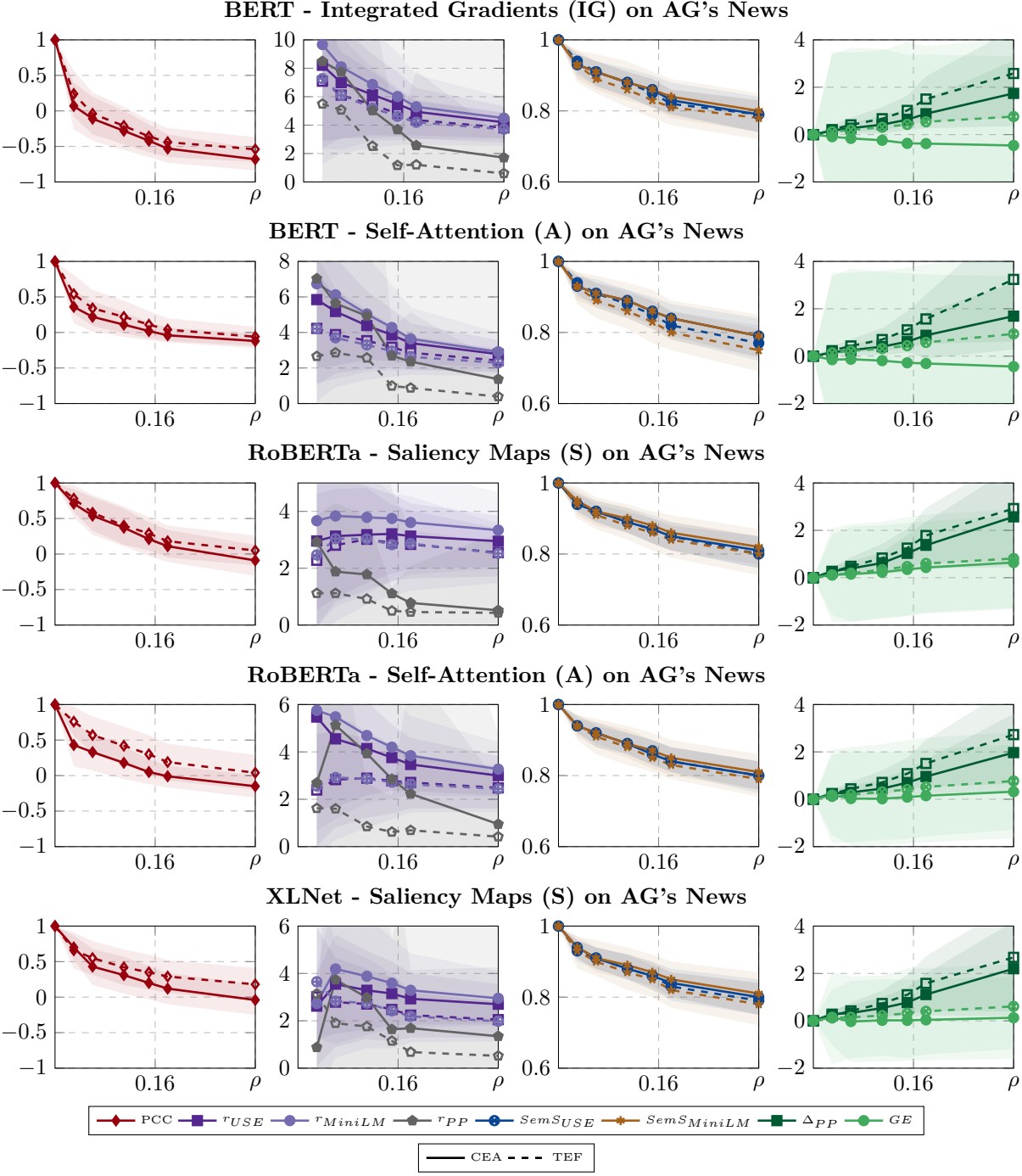

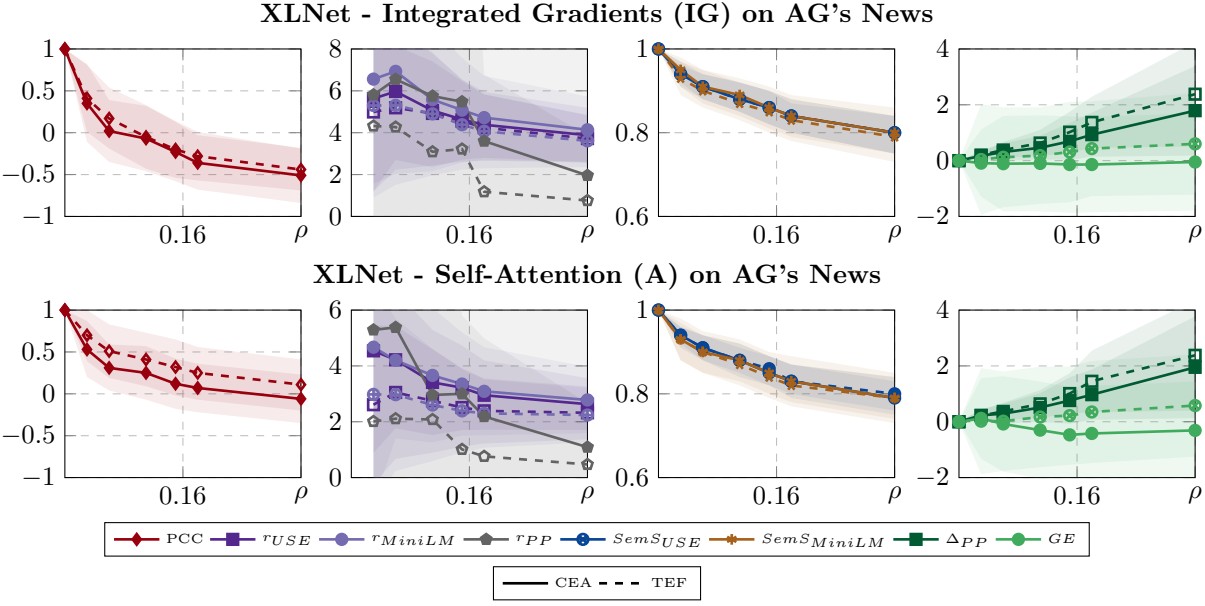

### A.5.2 MR

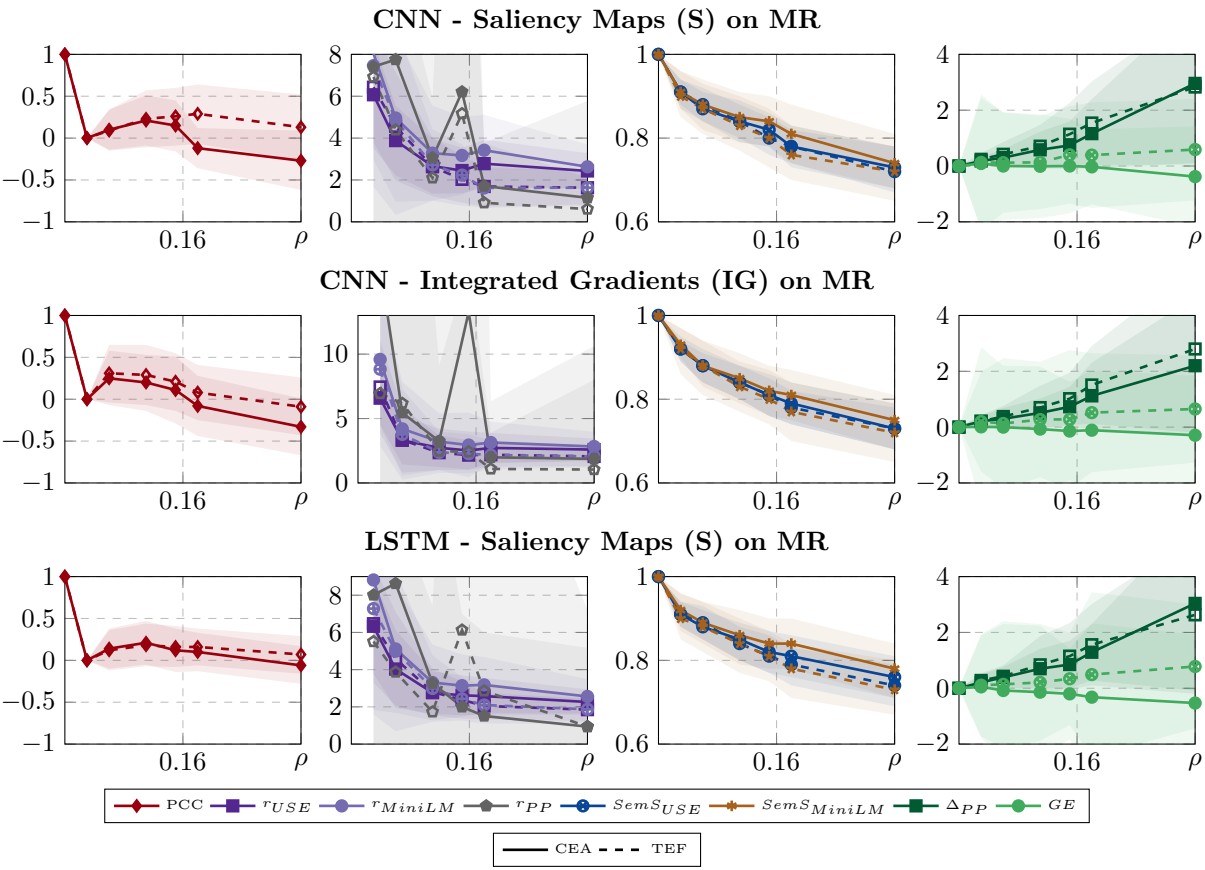

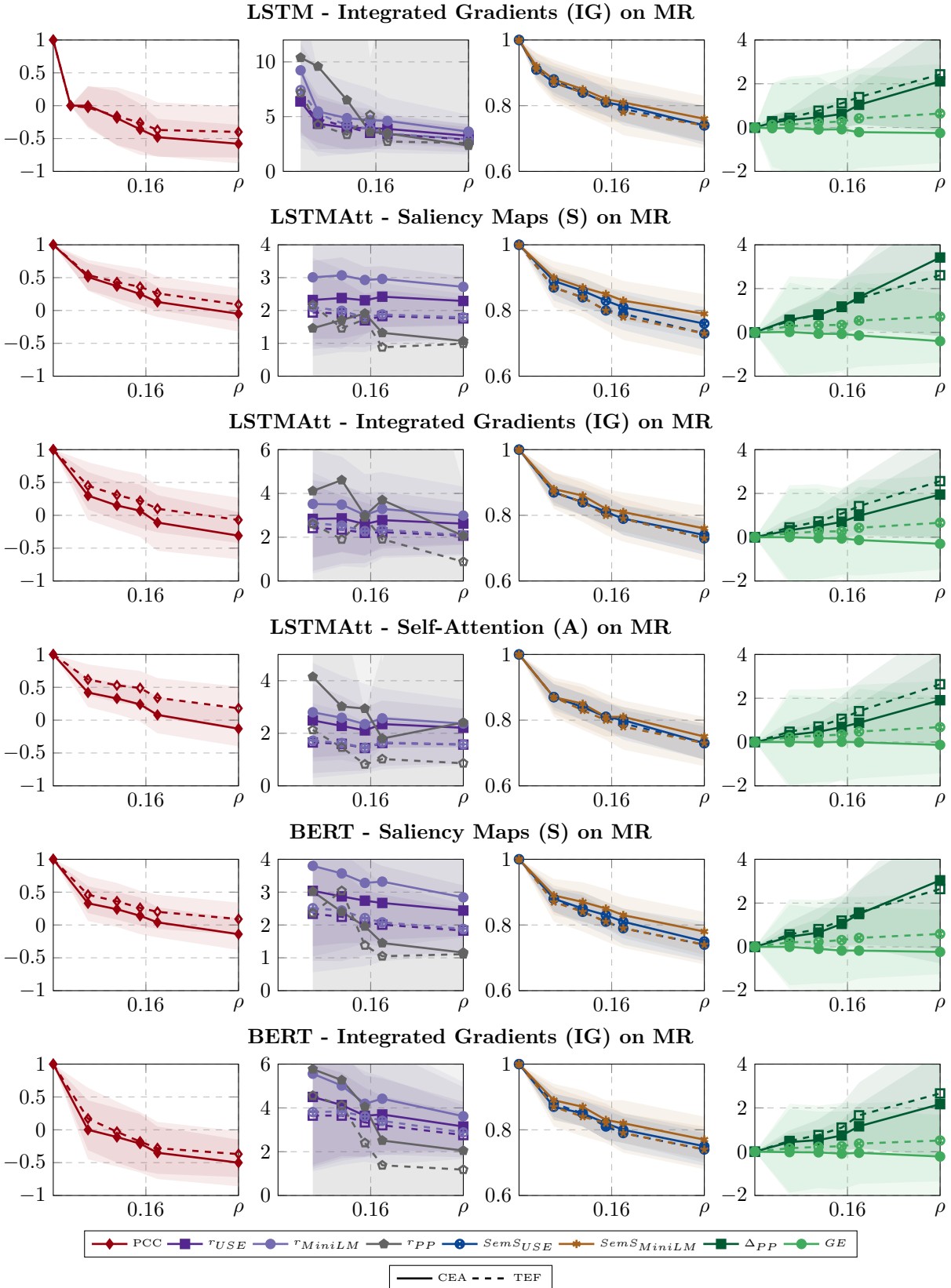

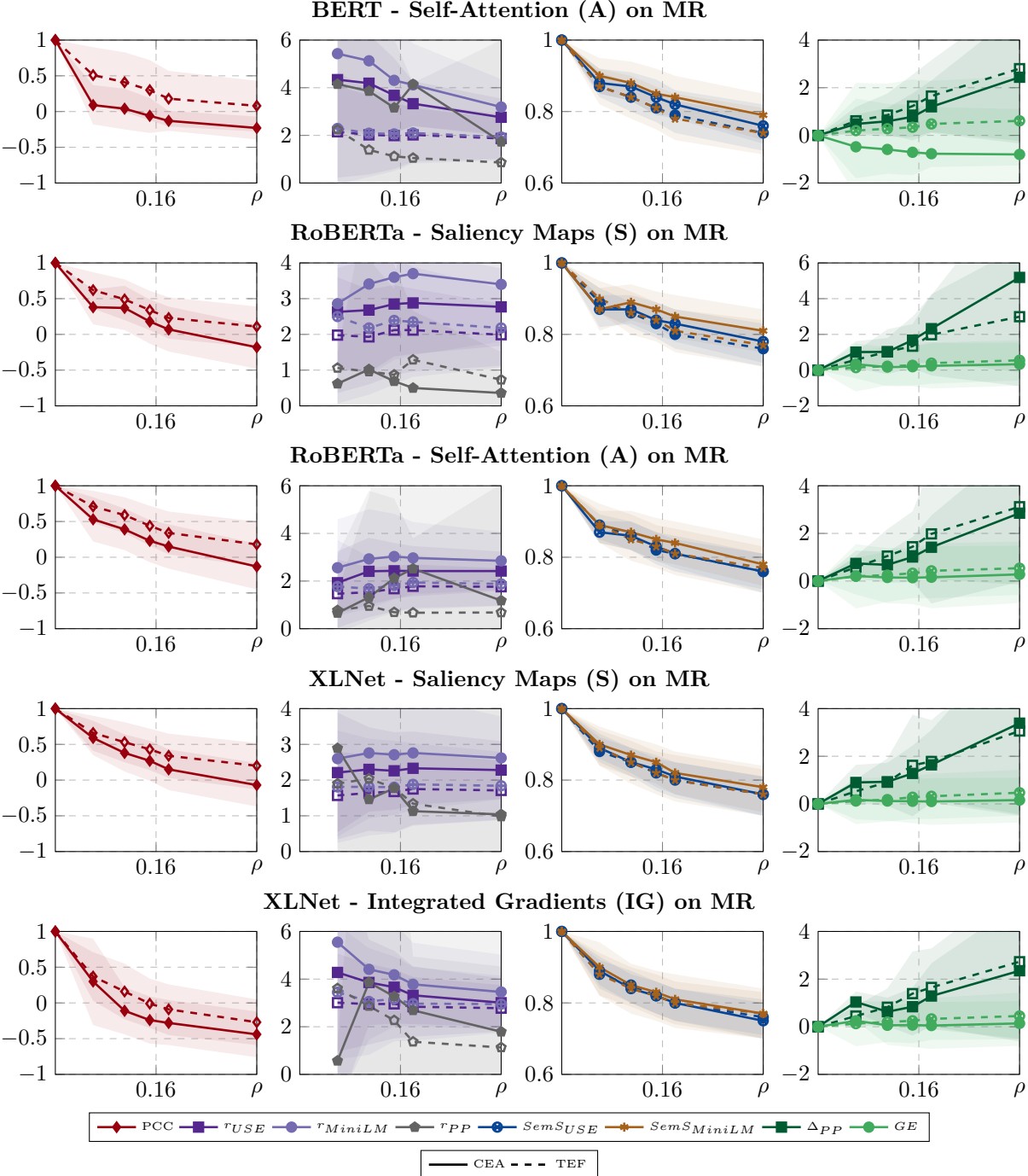

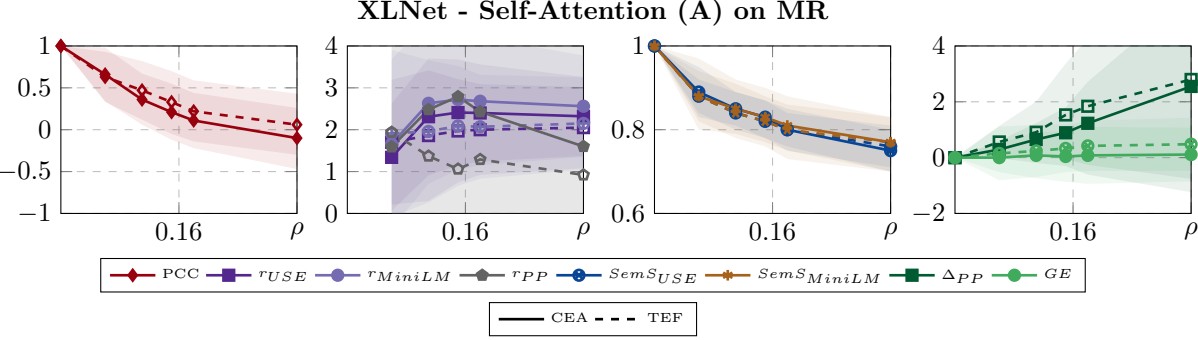

### A.5.3 IMDB

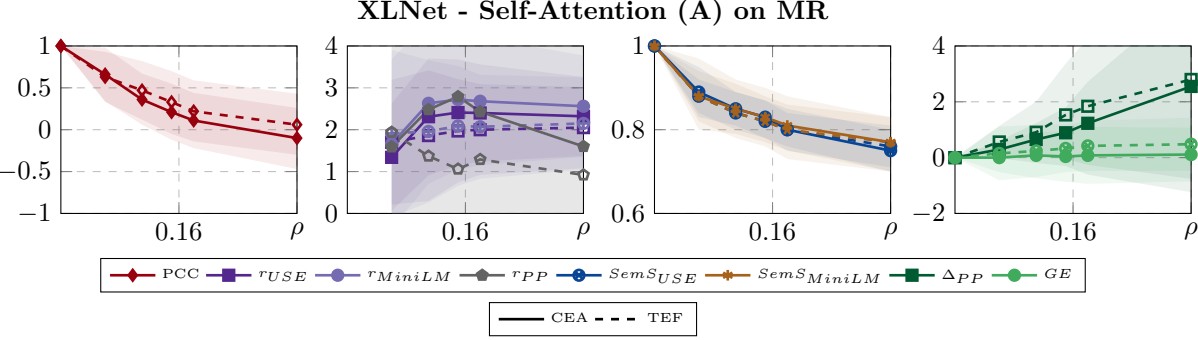

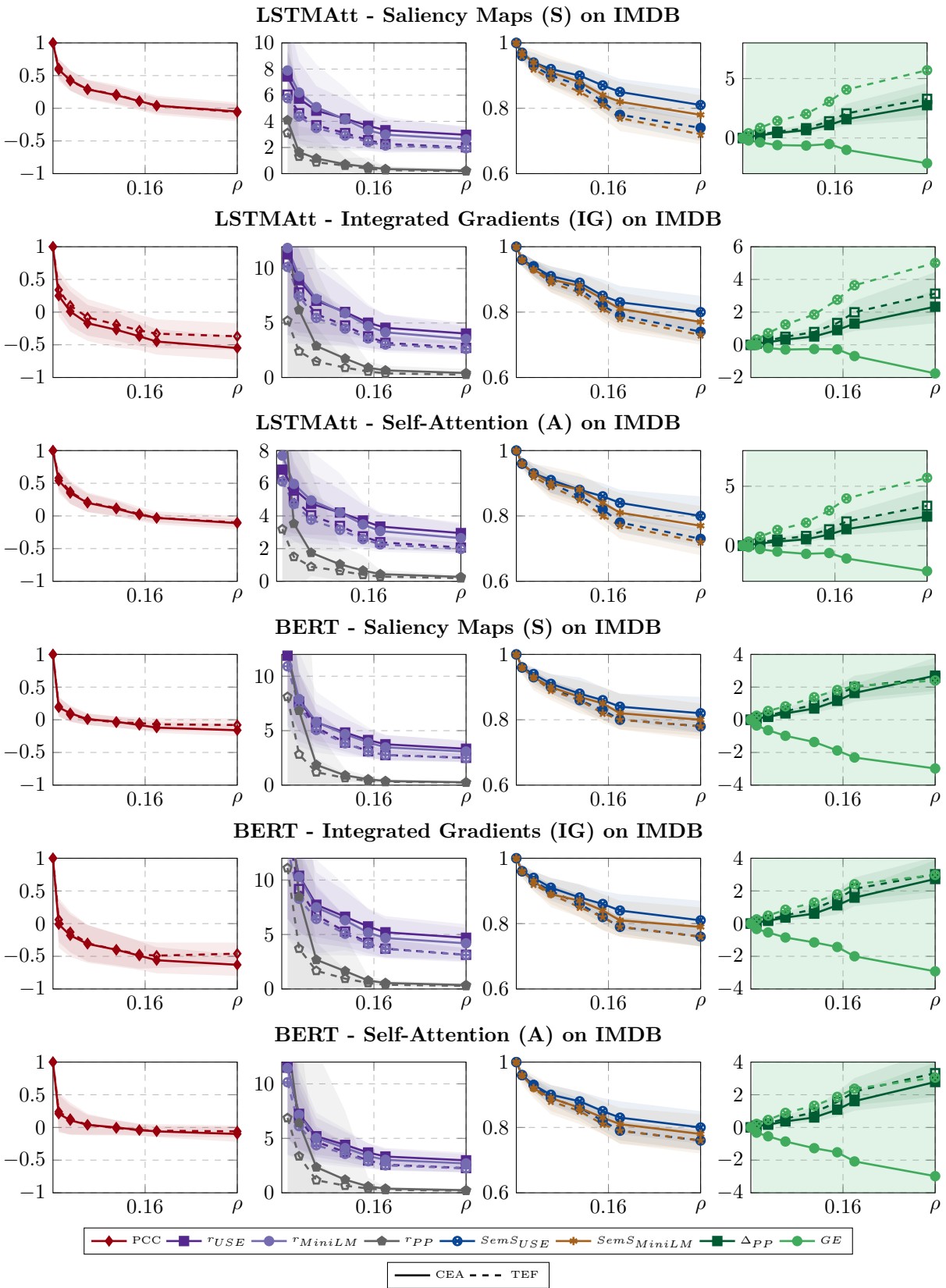

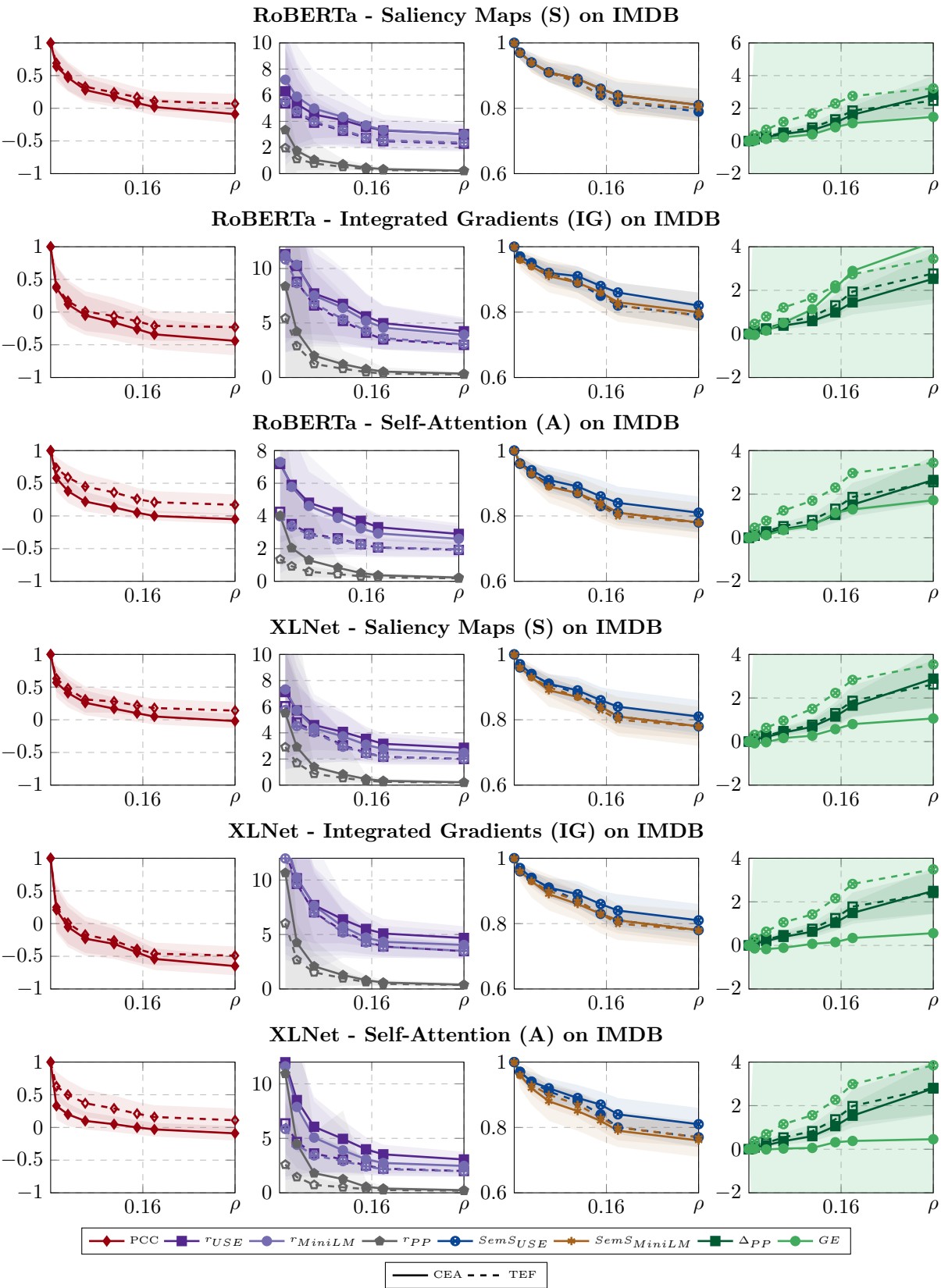

### A.5.4 Yelp

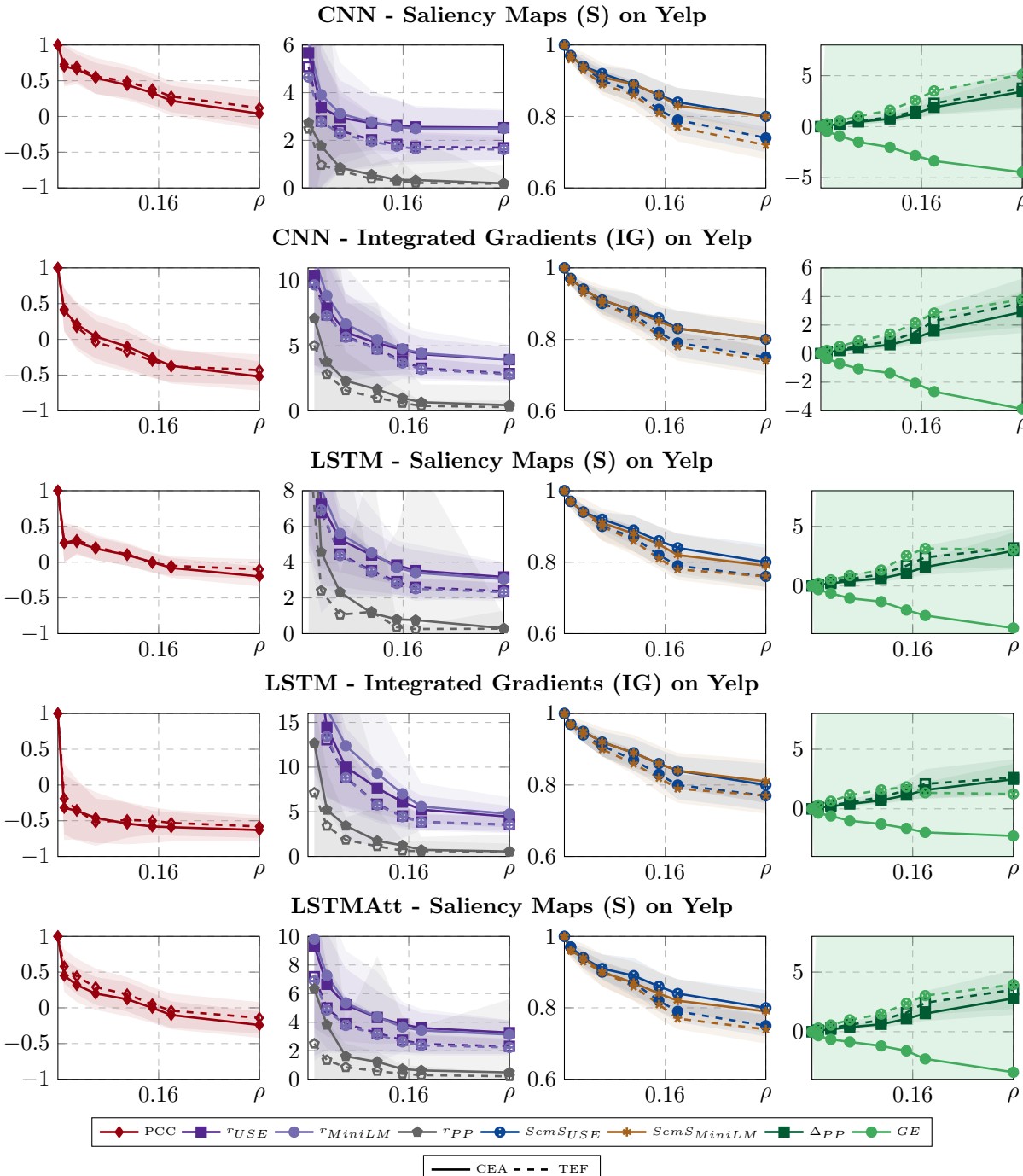

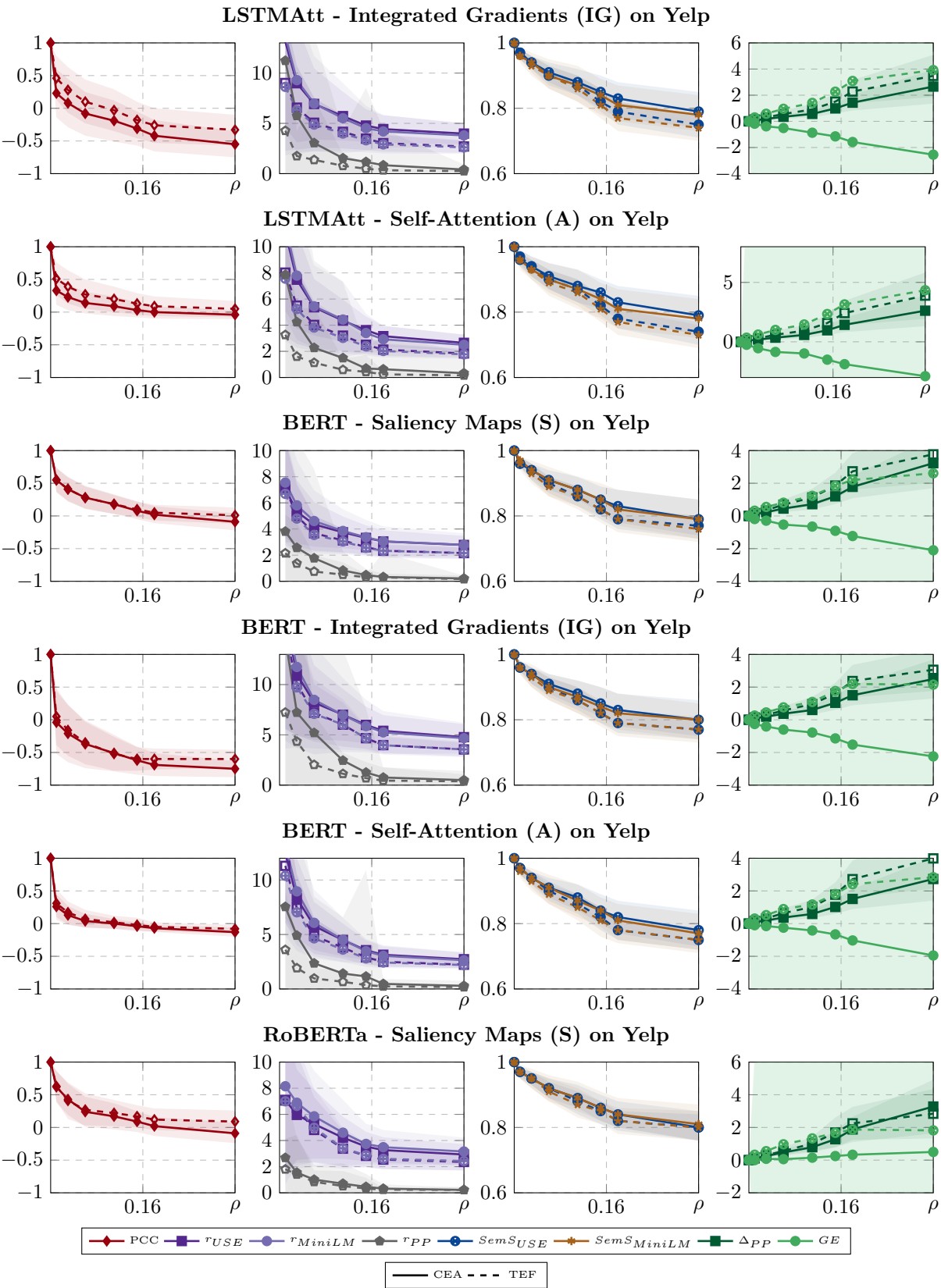

## RoBERTa - Integrated Gradients (IG) on Yelp

## RoBERTa - Self-Attention (A) on Yelp

## XLNet - Saliency Maps (S) on Yelp

## XLNet - Integrated Gradients (IG) on Yelp

### A.5.5   Fake News

## CNN - Saliency Maps (S) on Fake News

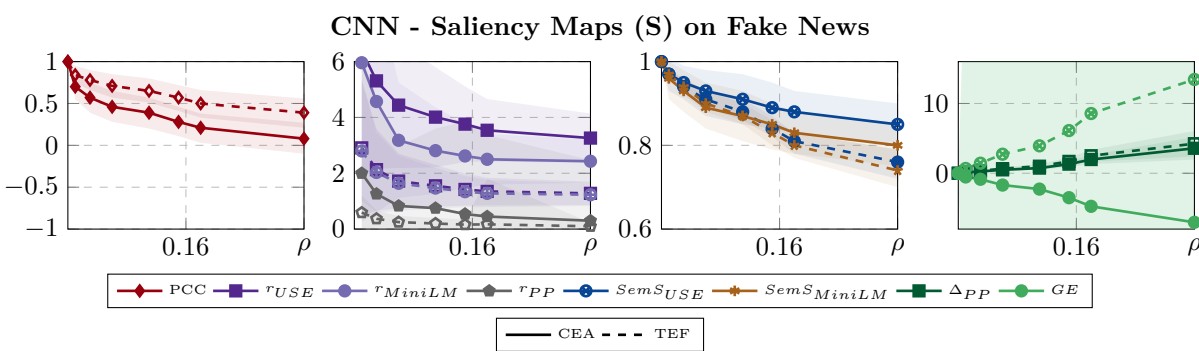

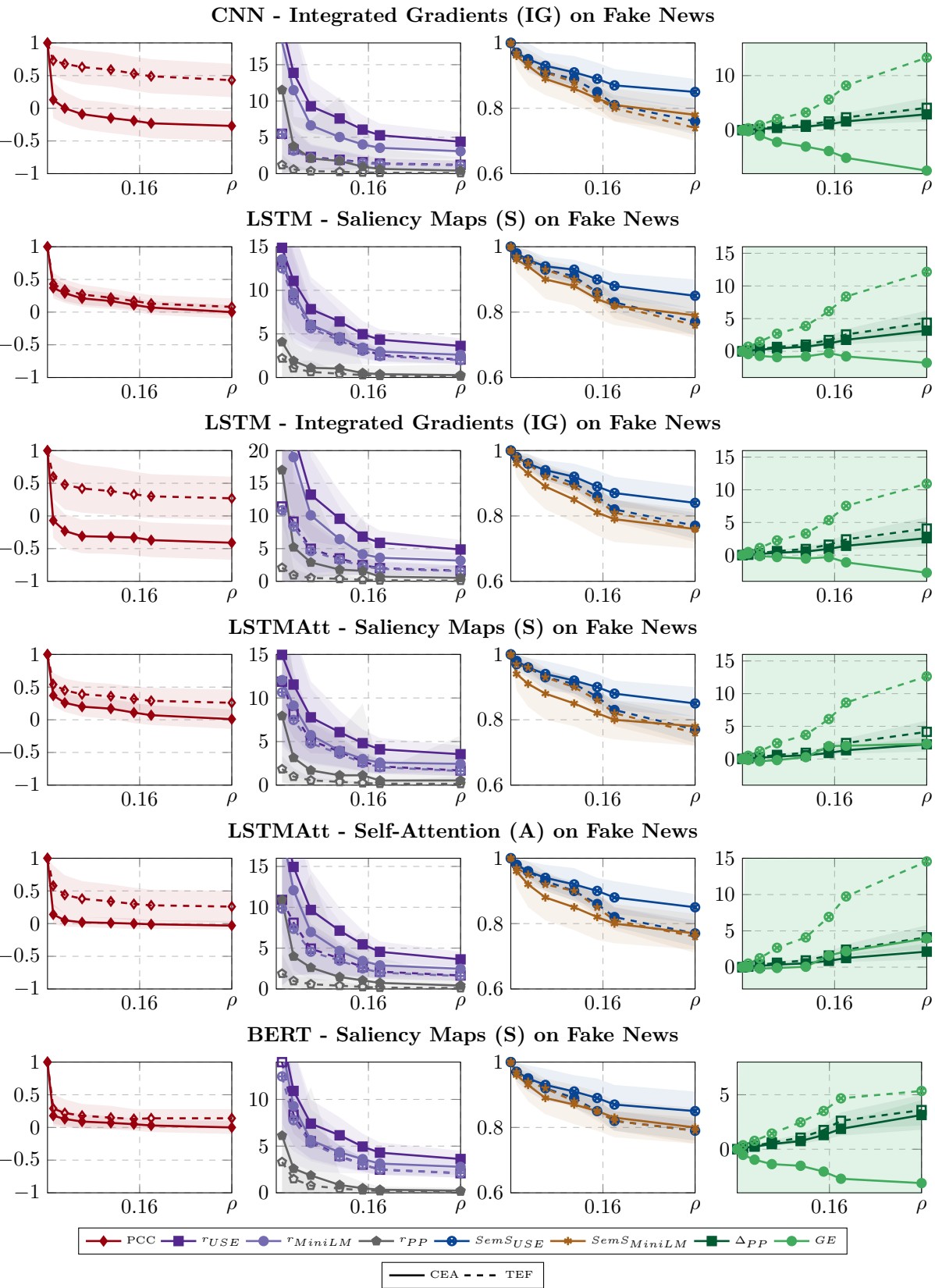

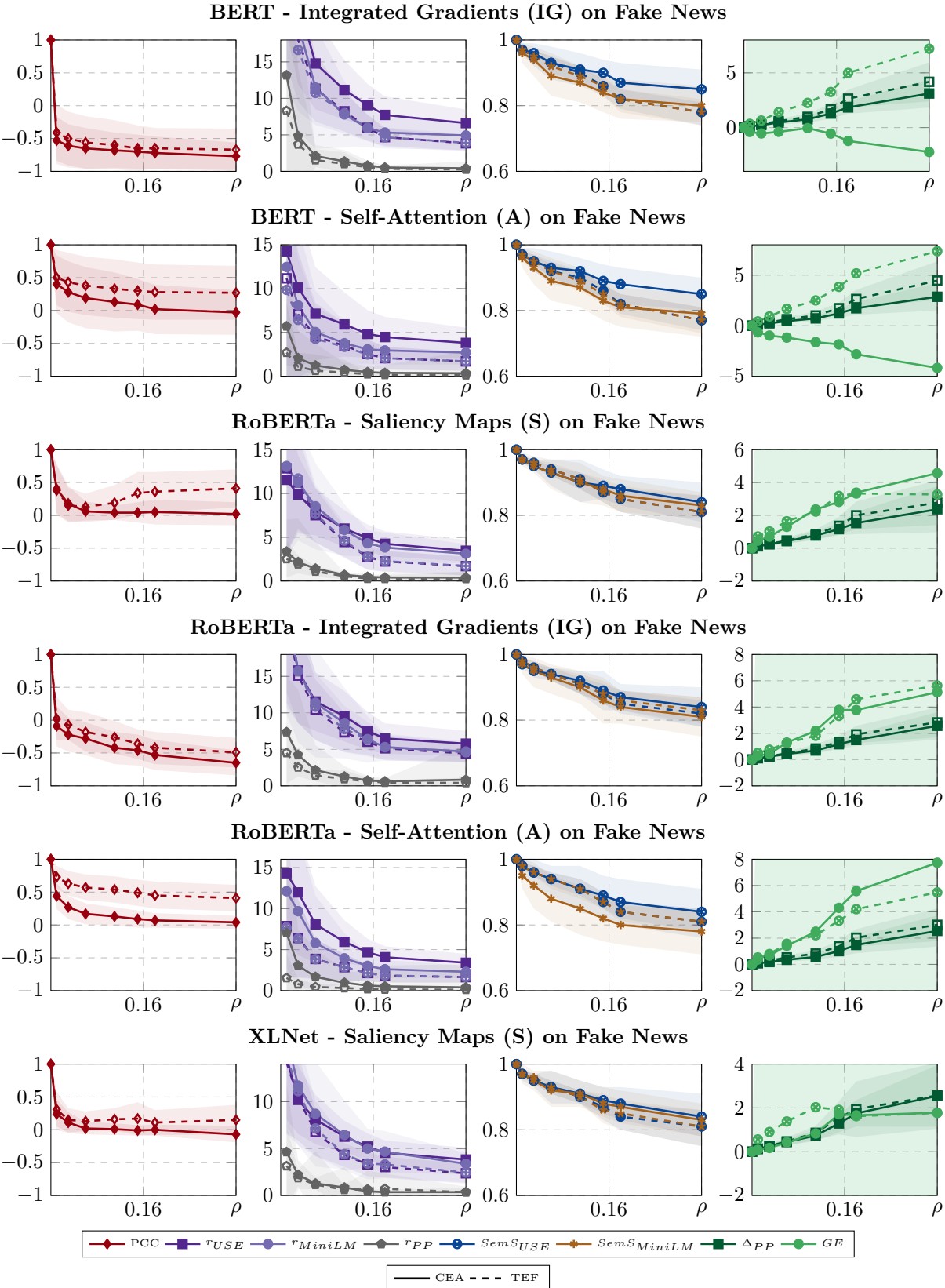

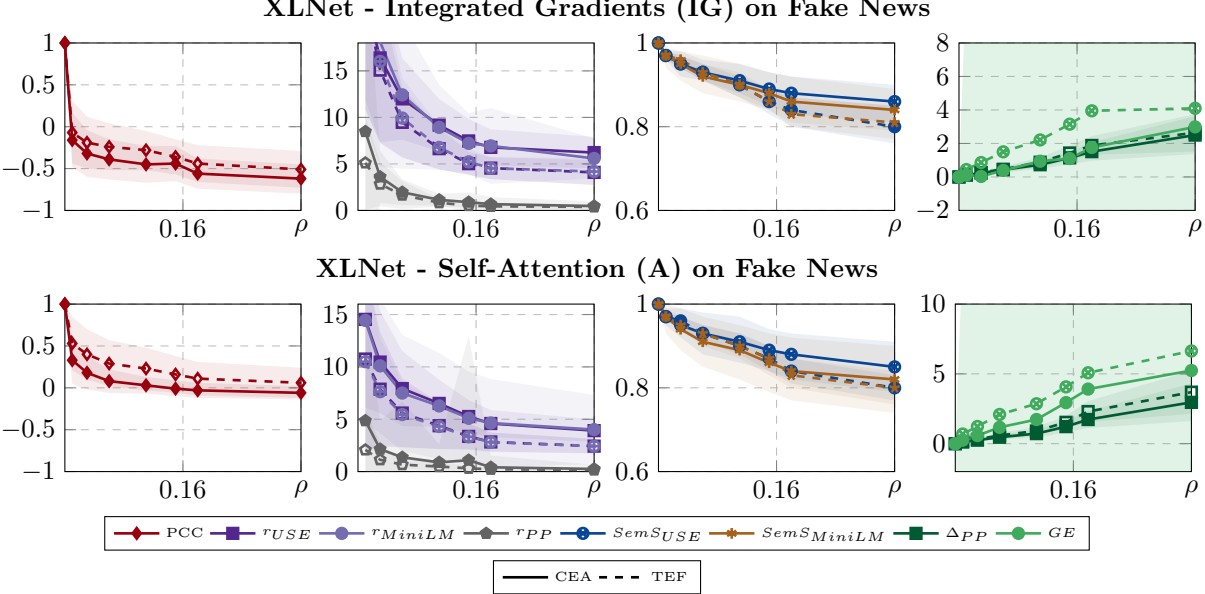

