# OpenReview forum: "Context-Aware Estimation of Attribution Robustness In Text"
_TMLR — Rejected by TMLR_

### Review · Reviewer_eiUT · 2023-10-06

**Summary Of Contributions:**

This paper introduce a new definition of attribution robustness in test classification. It uses semantic distance to detect perturbations in inputs. It also introduces a new attack called context-aware explanation attack which seems to work well using its evaluation metric.

**Audience:**

Yes

**Broader Impact Concerns:**

No broader impact concerns.

**Claims And Evidence:**

Yes

**Requested Changes:**

1. The authors should provide at least one experimental analysis to show their definition of AR is correlated with the failure cases appeared real-world applications. E.g., how does your AR metric correlates with the frequency where models fails to give correct explanations.
2. Refine the writing to explain what the metrics used in the experiment section indicate.

**Strengths And Weaknesses:**

The author propose a new measure an attribution robustness (AR) metric and a new attack method. The authors should provide at least one experimental analysis to show their definition of AR is correlated with the failure cases appeared real-world applications. E.g., how does your AR metric correlates with the frequency where models fails to give correct explanations. Besides, the writing is hard to follow for audience without the exact background.

---

> ### Author Response · Authors · 2023-11-11
> **Response on scope and metrics**
>
> Thanks a lot for the feedback.
>
> 1. The Reviewer raises an interesting question, which is the correlation between attribution robustness and other figure of metrics of explanations. In this work, we are agnostic on the quality of the attributions, such as whether they are plausible or faithful . We assume that the methods give faithful and relevant explanations (at least to some extent) and merely address their robustness. Nonetheless, the Reviewer's question hints at an important question that has been recently examined in the literature: that of quantifying robustness in relation to plausibility, based on the intuition that adversarial attacks that maintain the plausibility of the explanation are more dangerous than attacks that do not. As this point is outside the scope of our work (and as said we are interested to quantify robustness in isolation from other properties of the attribution), we add a short discussion related to this point in the conclusions and cite relevant literature. For completeness and to try to more directly address the Reviewer's request, we also provide a small experimental study on how the robustness correlates with a proxy for the quality of the explanations by computing our metric r for different explanations with increasing degree of corruption (obtained by partially and progressively randomizing the explanations. This can be found in the appendix, Figure 6, and shows that in the analyzed scenarios attributions tend to become less robust (and therefore r increases) as their degree of corruption is increased.
> 2. We refined the writing for the metrics and provide a clearer description in the re-submitted version.

---

### Review · Reviewer_LHfZ · 2023-10-12

**Summary Of Contributions:**

This paper proposes a new metric and a corresponding new method for evaluating the adversarial robustness of text attribution methods. The metric better captures imperceptibility of the attack by (a) considering the ratio of attribution-space change to input-space change, and (b) accounting for factors such as perplexity and grammatical correctness when measuring the change in input space. The result are more convincing adversarial examples that also fool attribution methods more reliably.

**Audience:**

Yes

**Claims And Evidence:**

Yes

**Requested Changes:**

See above

**Strengths And Weaknesses:**

The paper is very clearly written, and each component of the method is well-motivated, and to my knowledge, also novel. I think it is a great fit for TMLR. Based on the provided results, the method seems to be convincingly better than previous ones. A few more detailed comments:

- Based on the examples provided in the paper, the method indeed seems to be a big qualitative improvement over the TEF method---are these examples randomly selected? If not, there should be a set of randomly selected examples, a few in the main text and a more extensive list in the appendix.

- Related to the point above: there is some confounding introduced by the fact that the authors introduce both the metric and the corresponding method (in particular, it makes it somewhat unsurprising that the introduced method outperforms prior ones on a metric it is designed for)---this makes it important for the authors to either (a) include examples that let the reader see that the metric is reasonable; (b) show that their method outperforms prior ones on some other reasonable metric as well as the one proposed in this paper; or (c) run some kind of simple human evaluation that shows their metric is indeed more faithful to the intuitive notion of "imperceptibility."

- I did not understand from S4.2 how the three proposed distance metrics are combined, as they seem to be on completely different scales from one another

- I am similarly confused about Figure 2 - what determines if two lines appear in the same subfigure (e.g., why are PP and GE in the same subplot, but STS in a different subplot)? Also, all of the confidence intervals seem to overlap pretty strongly in this figure, which makes it hard to draw any conclusions---I think the authors should improve the clarity of this figure and also collect additional samples to make their point more strongly.

- Algorithm 1 makes sense but is very dense and hard to parse - the clarity could be improved with a few comments that let the reader follow along in plain English.

---

> ### Author Response · Authors · 2023-11-11
> **Response on some clarifications and changes**
>
> Thank you very much for the great feedback.
>
> 1. Yes indeed, the examples are manually picked so that they reflect the results the best. In order to address this, we added random samples in the text and appendix in the resubmitted version.
> 2. In fact most current work uses correlation metrics like Pearson’s coefficient (PCC) to compare AR of the models and explanations. In Figure 1, we report both our metric (r) as well as the PCC of original and adversarial explanations. Moreover, the semantic similarity between original and adversarial samples is also shown (SemS). Thus, we believe these are good indicators that show our method outperforming current attacks not only on our metric, but also other state-of-the-art metrics. We restrctured Figure 1 slightly to reflect that, and believe that this, together with additional random samples from the previous point should convincingly address this point.
> 3. We use the input distance metrics to capture imperceptibility of the perturbations applied to the input text. For the semantic distance (scaling from -1 to 1) as well as the relative increase in perplexity (Delta_PP), we compute different r-metrics according to Equation 2 and compare those for the architectures and attribution methods (two different metrics for two semantic embeddings - provided by the MiniLM and USE embeddings - these are r_USE and r_MiniLM in the plots, and one computed with the relative perplexity increase - r_PP). The relative perplexity increase indeed has a different scale than semantic embedding based distances, but the r-measures still can take values from 0 - +infinity for both cases. As for the grammatical errors, we do not compute r-values with that, but only compute the average number of grammatical errors in resulting adversarial samples as function of rho.
> 4. The main determining factor is what values the metrics can take. The leftmost plot contains only the average PCC between original and adversarial attributions, taking values between -1 and 1. The second plot from the left contains all r-values, ranging from 0 - +infinity. The third from the left has the two average semantic similarities, computed with USE and MiniLM embeddings. These can take values from 0-1, the plot is capped though from 0.6 - 1. The rightmost plot contains the relative perplexity increase and average number of grammatical error, taking values from -infinity to +infinity (theoretically). We thank the Reviewer for pointing out this possible source of confusion. We made this distinction clearer in the re-submitted version of the paper by potting the headers and more details.
> 5. Thank you for the suggestion to improve the readability of Algorithm 1. We added comments to help clarify its steps in plain language.

---

### Review · Reviewer_tRXZ · 2023-10-20

**Summary Of Contributions:**

This work studies the problem of attribution robustness for text classification. First, they propose a new definition of attribution robustness that considers both the distance between the original and the perturbed text, as well as the semantic similarity. This definition further requires the perturbed input to share the same prediction label and similar semantic meaning with the original input, which makes the attack harder to detect. Based on the new attribution robustness measurement, this work further proposes a new attack called context-aware explanation attack (CEA), with the following changes compared to the baseline TEF attack: (1) for each token, the substitution candidates are extracted from masked language models instead of separate synonym embeddings; and (2) batch masking to reduce the computational cost for attacks. They compare their CEA attack to TEF on a range of text classification tasks with a couple of models including CNN, LSTM, and Transformer-based architectures. They demonstrate that CEA generally outperforms TEF and generates more fluent adversarial input with a higher semantic similarity,  and batch masking effectively reduces the runtime.

**Audience:**

Yes

**Broader Impact Concerns:**

No concern on the ethical implications of the work.

**Claims And Evidence:**

Yes

**Requested Changes:**

The submission will benefit from more clarifications regarding the novelty and significance of the proposed attack, especially highlighting the differences to prior methods.

1. In the method description, clarify which part is from prior work, and what are the new techniques proposed in this work.

2. Add ablation studies to justify the importance of design choices for the new attack.

3. Explain the implementation details of MLM for candidate selection.

**Strengths And Weaknesses:**

Strengths:

1. Attribution robustness is an interesting topic, and this is an important measurement to improve the trustworthiness of prediction results.

2. The experiments cover different metrics to measure attributions and semantic similarity, and show that the proposed attack outperforms the baseline across the board.

Weaknesses:

In general, I think the paper lacks the clarity on the novelty and significance of the proposed approach.

1. In the method description, the writing does not clearly specify which part is from prior work, and what are the new techniques proposed in this work. For example, this work proposes a new definition of attribution robustness, but the paper does not clearly state the differences to prior definitions, the benefits of the changes, and how this new definition affects the final attack quantitative performance. More explanation is needed to enable a better understanding.

2. In terms of the approach, I think the main new techniques include candidate selection and batch masking. However, these techniques are commonly used in prior work and are not technically novel. Also, the evaluation lacks the ablation studies solely on the effect of MLM for candidate selection: they only compare to TEF which can contain other differences in implementation.

3. The implementation details of MLM for candidate selection is unclear. Is the candidate set filtered from the full vocabulary for each token, or is there a pre-filtered set per token before applying MLM to select the final candidate set?

4. Although the comparison to the baseline shows improvement on multiple metrics, in general the significance of the improvement is unclear, and in many cases the increase seems marginal.

---

> ### Author Response · Authors · 2023-11-11
> **Response on clarifications and ablations**
>
> Thank you for the thorough response, we think that the mentioned points are very valid.
>
> 1. We clarified the contributions and existing prior work in the methods description, focusing on Section 4. You are right that some of the used methods, like batch masking and the candidate selection with MLMs are not new concepts introduced by us, however, we are the first to apply these in the setting of attribution robustness estimation. No other work has done so, to the best of our knowledge.
> 2. As requested by the reviewer, we ran an ablation study to quantify the contribution of the novel elements of our attack. In particular, we investigate how Step 2 (Candidate selection) influences our results by randomising it and comparing the results to CEA. We also ran an even stronger ablation by completely removing the MLM in the candidate selection step, which results in an algorithm that is equivalent to TEF. We report these results in the new version of the paper, comparing them with the original unablated algorithm in the new figure 5 in the re-submitted version of the paper.
> 3. We added a more detailed description of how we use MLMs for candidate selection in Section 4.3.
>
> We are confident that these additional experiments and addition to the paper should address the valid concerns pointed out by the Reviewer, and we're grateful for the useful feedback that resulted in these improvements of our manuscript.

---

### Decision · Action_Editor_5kbR · 2023-12-10

**Recommendation:** Reject

**Comment:**

This paper has mixed opinions. I also share the concerns of the reviewers. The most serious concern is the high similarity between this paper and the TEF paper. After reading the revised text, reviewers' concerns, and the previous paper, I think this paper needs a minor revision. Most of the techniques used in this paper are not newly introduced, but this paper can be misread in that this paper first proposes (1) the concept of AR, (2) using PCC for AR, (3) word importance ranking, Batch Masking, and Candidate Extraction -- it is clarified in 5.2.3., but a reader can misunderstand it in Section 4. In my opinion, the contribution regarding the candidate extraction by MLM can also be confusing to the readers. This is because this technique itself is not new and widely used, as pointed out by Reviewer tRXZ. As far as I understood, the technical contribution of this paper is to employ the candidate extraction by MLM to the TEF attack. However, a reader can be confused that this technique itself is newly introduced by this paper.

Therefore, this paper should clarify the contribution of TEF and this paper. For example, "We are the first to introduce a definition of attribution robustness (AR) in text classification that takes both the attribution distance and perceptibility of perturbations into account" can confuse the readers that AR is first introduced in this paper. However, in fact, AR is already proposed in the TEF paper, but without the division term defined by the sentence similarity. The overall attack algorithm is also highly similar to the TEF attack. Although it is now clarified in the revised paper, I think it should be clarified in earlier sections, such as Section 4.3 or even Section 1. Similarly, the last contribution, "We successfully speed up robustness estimation with the usage of distilled language models and batch masking" can also be confusing because MLM and batch masking are not newly introduced techniques. This paper employs the techniques for TEF.

Another problem is the novelty of this work. As far as I understood, the technical difference between this paper and TEF is

1. Slightly different AR definition (using an additional sentence distance in PPL)
2. Using a different candidate selection strategy: TEF uses synonym embeddings, and CEA uses MLM.

TMLR evaluation criteria request "Unlike many other journals, TMLR only accepts original contributions that ..." and "claims are supported by accurate, convincing and clear evidence". I think the current paper's claim does not satisfy this criterion because the main claim is highly duplicated to TER and previous works (e.g., MLM candidate strategy). I think this paper should be rewritten to clarify its main claim and its original contribution compared to the other works.

I also considered "minor revision" because I think that the technical findings and contributions of this paper are somewhat acceptable to TMLR if the contribution and the originality are clarified. However, I think the modification will be heavy for satisfying the originality criteria and the revised manuscript will need another review round. Therefore, I recommend reject with a major revision. Please revise the main paper for clarifying the technical novelty and the originality.

**Audience:**

I believe that the adversarial robustness of text classifiers can be an interesting topic in this area. I think this direction is worth exploring.

**Claims And Evidence:**

According to the manuscript, this paper is the first to introduce a new robust metric named "Attribution Robustness (AR)". AR is defined by the worst distance of an attribution function (e.g., saliency) between the modified text (where the modified text is limited to its neighborhoods defined by an attribution function) and the previous text. In other words, AR is the worst-case distance while preserving its original prediction, like the certified robustness. In the paper, the Pearson Correlation Coefficient (PCC) is used for the attribution function distance, and perplexity is used for the sentence distance.

However, unlike the manuscript's explanation, AR is already defined by TEF (Ivankay et al., 2022). The formulation is almost similar, but the only difference is that AR in this paper is divided by sentence distance, while TEF's is not.

Using the metric, this paper proposes a new black-box attack, named Context-AwareExplanationAttack (CEA). Due to the intractable search space for the token perturbation, this paper employs widely-used simplification (Li et al. 2020), e.g., restricting the token substitutions from the predefined vocabulary. After the candidate tokens for substitution are selected, an MLM predicts the plausible alternatives. To reduce the inference time, CEA employs a batch masking strategy, resulting a comparable running time to TEF despite using an extra MLM. As Reviewer tRXZ pointed out, this process is highly similar to the TEF (Ivankay et al., 2022). The main difference between TEF and CEA is the selection of the alternative tokens: TEF selects a token from the synonym embeddings (Mrkšic et al. 2016), while CEA uses an MLM (distilled BERT) for estimating the alternative tokens.

In the experiments, the proposed CEA consistently shows better results than the baseline TextExplanationFooler (TEF) attack in the proposed metric, Attribution Robustness.

Overall, I think the main focus of this paper is to tackle the implausible attacked texts by TEF. However, this paper still has a high similarity to the TEF paper and can confuse the readers between the original contribution of this paper and the contribution of TEF.

**Resubmission Of Major Revision:**

The authors may consider submitting a major revision at a later time.

---

> ### Author Response · Authors · 2023-12-13
>
> Dear Action Editor and dear Reviewers,
>
> Thank you very much for your time and effort in carefully reviewing our paper and assessing its fit to TMLR. We are very grateful for the encouraging comments and for the clear analysis of the weaknesses of our work. We will evaluate the suggestion of resubmitting the paper after major revisions to address the provided comments.
> In the meantime, we'd like to thank you again for the constructive criticism.
>
> Best regards,
> TMLR Paper1588 Authors